# NK cells are activated and primed for skin-homing during acute dengue virus infection in humans

Christine L. Zimmer[1], Martin Cornillet[1], Carles Solà-Riera[1], Ka-Wai Cheung[2], Martin A. Ivarsson[1], Mei Qiu Lim[2], Nicole Marquardt [1], Yee-Sin Leo[3,4,5,6], David Chien Lye [3,5,6], Jonas Klingström [1], Paul A. MacAry[7], Hans-Gustaf Ljunggren[1], Laura Rivino[2,8,9] & Niklas K. Björkström [1,9]

Despite animal models showing that natural killer (NK) cells are important players in the early defense against many viral infections, the NK cell response is poorly understood in humans. Here we analyze the phenotype, temporal dynamics, regulation and trafficking of NK cells in a patient cohort with acute dengue virus infection. NK cells are robustly activated and proliferate during the first week after symptom debut. Increased IL-18 levels in plasma and in induced skin blisters of DENV-infected patients, as well as concomitant signaling downstream of the IL-18R, suggests an IL-18-dependent mechanism in driving the proliferative NK cell response. Responding NK cells have a less mature phenotype and a distinct chemokine-receptor imprint indicative of skin-homing. A corresponding NK cell subset can be localized to skin early during acute infection. These data provide evidence of an IL-18-driven NK cell proliferation and priming for skin-homing during an acute viral infection in humans.

[1] Department of Medicine Huddinge, Center for Infectious Medicine, Karolinska Institutet, Karolinska University Hospital, Stockholm, Sweden. [2] Programme in Emerging Infectious Diseases, DUKE-NUS Medical School, Singapore, Singapore. [3] Institute of Infectious Diseases and Epidemiology, Communicable Disease Centre, Tan Tock Seng Hospital, Singapore, Singapore. [4] Saw Swee Hock School of Public Health, National University of Singapore, Singapore, Singapore. [5] Yong Loo Lin School of Medicine, National University of Singapore, Singapore, Singapore. [6] Lee Kong Chian School of Medicine, Nanyang Technological University, Singapore, Singapore. [7] Immunology Programme, Life Science Institute and Department of Microbiology and Immunology, National University of Singapore, Singapore, Singapore. [8] School of Cellular and Molecular Medicine, University of Bristol, Bristol, UK. [9]These authors contributed equally: Laura Rivino, Niklas K. Björkström. Correspondence and requests for materials should be addressed to N.K.B. (email: niklas.bjorkstrom@ki.se)

Natural killer (NK) cells are innate lymphocytes important in the early immune response against many viral infections[1–3]. From experimental model systems, it is known that NK cells participate in the control of acute infections by many different viruses[3,4]. In humans, their importance is underscored in rare NK cell deficiencies, resulting in a high susceptibility to many viral and other infections[5]. Furthermore, recent work has shown that human NK cells are activated and/or expanded during numerous viral infections[6–12]. However, many details are missing with respect to how an early NK cell response to an acute viral infection is driven in humans.

Human NK cells are divided into two phenotypically and functionally distinct subsets based on their expression of CD56: CD56[bright] and CD56[dim] NK cells. CD56[bright] NK cells are prominent in peripheral organs where they primarily respond to cytokines and exhibit immunoregulatory functions by virtue of a profound cytokine and chemokine-producing capacity. CD56[dim] NK cells are prone to target cell interaction and, as a consequence, have high cytotoxic as well as cytokine-producing capacities, and constitute the dominant NK cell population in peripheral blood[1,13]. NK cell activation is regulated by signals from activating and inhibitory receptors, as well as cytokine and chemokine receptors[3,13,14]. A process referred to as NK cell education further modulates their responsiveness, rendering NK cells more functionally potent to missing self. Education occurs predominantly via interactions between inhibitory KIRs and NKG2A, and their respective cognate histocompatibility leukocyte antigen (HLA) class I ligands[15,16]. In addition to education, NK cells undergo a regulated differentiation process during which both phenotypic and functional changes occur[17].

Early in infection, NK cells are activated by cytokines such as type I interferons, interleukin-12 (IL-12), IL-15, and IL-18[1–3]. This activation may initiate changes in cell signaling pathways, which ultimately impact NK cell effector responses[18,19]. These include directed killing of infected cells via death receptors or directed release of cytotoxic granules containing perforin and granzymes, as well as the production and release of anti-viral cytokines[13,20]. However, pathogen clearance also requires rapid recruitment of immune cells to target organs. Over the past decade, progress has been made in the development of experimental model systems elucidating homing mechanisms for NK cells during infections in mice[21,22]. These studies have revealed the importance of distinct chemokine–chemokine receptor pairs in NK cell recruitment to liver, lung, gut, and central nervous system[23–29]. However, little is known about NK cell trafficking to peripheral organs in humans during acute viral infections.

Dengue virus (DENV) infection is a mosquito-borne viral disease annually affecting up to 400 million individuals worldwide[30]. Here, we perform an in-depth evaluation of the early NK cell response following DENV infection in a cohort of patients suffering from dengue fever (DF). Patients with acute DENV infection are recruited and followed from the onset of clinical symptoms until resolution of disease and beyond. Using a high-dimensional unsupervised analysis approach, we comprehensively analyze the NK cell response. The results show that both CD56[bright] and a subpopulation of CD56[dim] NK cells respond vigorously to the acute infection. This occurs irrespectively of their educational status and no evidence for specific adaptive-like NK cell responses is noted. Increased levels of plasma and skin IL-18 are observed during acute infection and are shown to induce signaling downstream of the IL-18R in NK cells, suggesting a pivotal role for IL-18 in the induction of the NK cell responses. Finally, the CD56[bright] NK cell population expresses chemokine receptors associated with homing to organs commonly affected during the natural course of clinical DENV infection including the skin. In conclusion, our results provide novel insights into the dynamics and specifics of the early NK cell response during an acute viral infection in humans.

## Results

**NK cells are robustly activated during acute DENV infection.** To investigate the NK cell-mediated immune response during acute DENV infection, 32 patients were followed longitudinally from an early stage of infection, after symptom debut to the post-febrile phase and into convalescence. All, except for one, of the included patients presented with DF, a normal presentation of acute DENV infection. Detailed clinical characteristics of the cohort are described in Supplementary Table 1. First, we analyzed the NK cell response in the patients throughout the course of infection (Fig. 1a–d, Supplementary Fig. 1a–c). A high percentage of both CD56[bright] and CD56[dim] NK cells were stained positive for the proliferation marker Ki67 at the acute phase of infection (Fig. 1a, b). The frequency of Ki67-expressing NK cells subsequently declined throughout the post-febrile and convalescent phases. A similar pattern of activation marker expression was seen for CD69 as well as for CD38, the latter depicted as median fluorescence intensity (MFI), albeit with lower percentages of CD69 expression on CD56[bright] NK cells during the acute phase of infection (Fig. 1c, d). Of note, the expression of Ki67 and CD69 was largely mutually exclusive. CD56[dim] NK cells are known to undergo a regulated differentiation process[17]. No remarkable changes in frequencies of NK cells in peripheral blood were observed throughout the infection in DENV-infected patients (Supplementary Fig. 1b, c). To further characterize the responding CD56[dim] NK cells during the acute phase of infection, Ki67 expression was assessed in relation to distinct CD56[dim] NK cell differentiation stages[17]. Notably, responding cells were predominantly observed among less mature CD57[−] cells, while more mature CD57[+] cells showed a markedly lower response in terms of activation (Fig. 1e, f, Supplementary Fig. 1d, e). Taken together, immature NK cells were robustly activated during acute DENV infection. This prompted us to perform a more in-depth analysis of the responding cells.

**Responding NK cells exhibit an immature phenotype.** The ability to simultaneously compare the detailed phenotype of responding and non-responding NK cells in multiple patients can provide a better understanding of the exact nature of the NK cell response observed. For this purpose, Flow Cytometry Standard (FCS) files containing populations of responding (Ki67[+]) and non-responding (Ki67[−]) CD56[bright] and CD56[dim] NK cells, respectively, from patients with acute infection were assigned electronic barcodes. The FCS files were concatenated and analyzed with stochastic neighbor embedding (SNE). SNE is a dimensionality reduction algorithm that generates two-dimensional representations of flow cytometry data at the single-cell level while preserving local and global geometry of the data[31,32]. Eleven distinct NK cell receptors, simultaneously stained for, were selected for clustering and two-dimensional maps of responding and non-responding NK cells were constructed (Fig. 2a–d). By generation of residual plots, considerable differences in multi-dimensional space between responding and non-responding CD56[bright] and CD56[dim] NK cells were revealed (Fig. 2a, c).

Based on the obtained SNE maps, expression intensities of the 11 parameters within the clusters of differentially abundant cells were evaluated (Fig. 2b, d). This revealed responding CD56[bright] NK cells to show only minor differences in phenotype compared to non-responding CD56[bright] NK cells (Fig. 2b). On the other hand, responding CD56[dim] NK cells displayed a clearly less

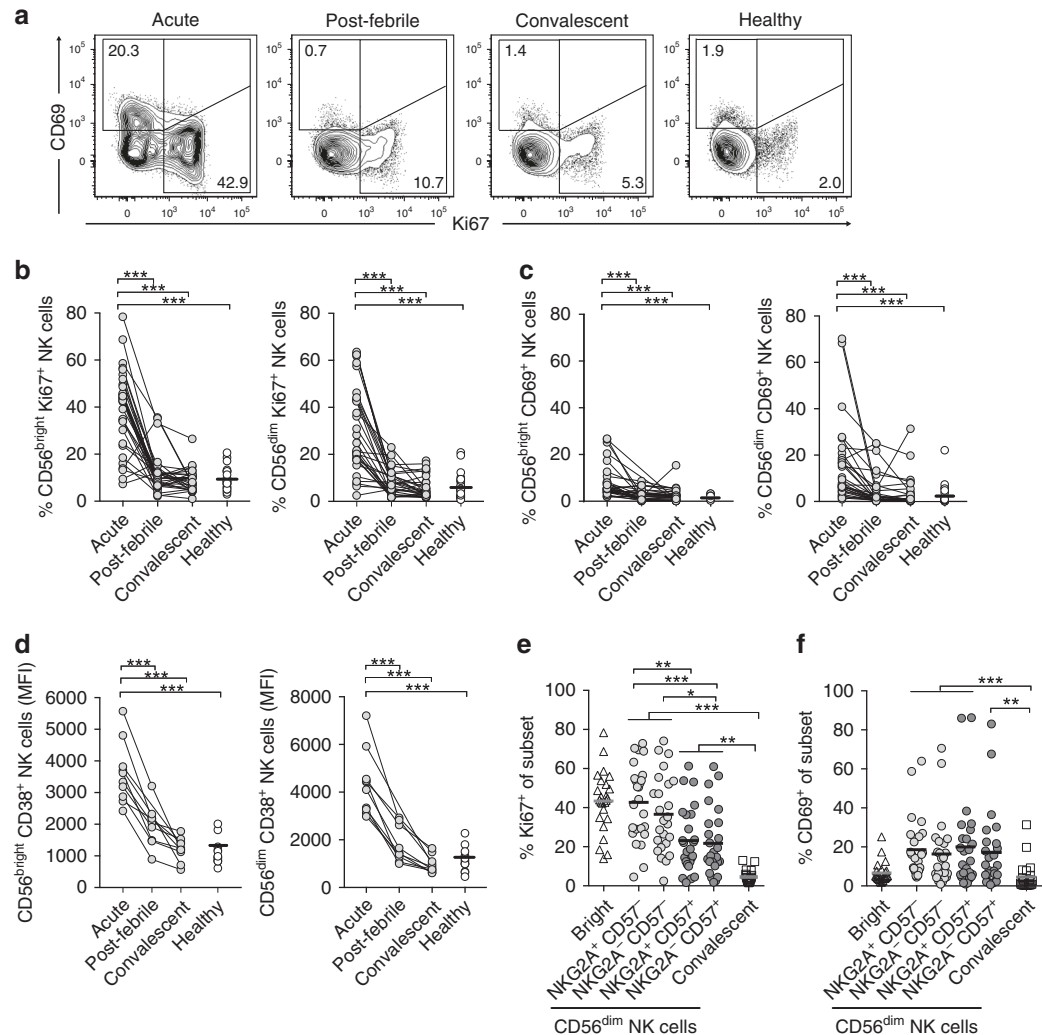

**Fig. 1** Natural killer (NK) cells are robustly activated during the acute phase of dengue virus (DENV) infection. **a** Representative staining for CD69 and Ki67 expression on NK cells from one representative patient during the acute, post-febrile, and convalescent phase of DENV infection, and one healthy control. Summary of data showing **b** Ki67, **c** CD69 (acute, $n = 31$; post-febrile, $n = 29$; convalescent, $n = 29$; healthy control, $n = 26$), and **d** CD38 (acute, $n = 10$; post-febrile, $n = 9$; convalescent, $n = 10$; healthy control, $n = 13$) expression on CD56$^{bright}$ and CD56$^{dim}$ NK cells shown as median fluorescence intensity (MFI), respectively. **e, f** Summary of data on **e** Ki67 and **f** CD69 expression on CD56$^{bright}$ as well as CD56$^{dim}$ NK cell subsets, respectively. The CD56$^{dim}$ NK cells are stratified by differential expression of NKG2A and CD57. NK cells from DENV-infected patients at the acute ($n = 25$) and convalescent ($n = 23$) phases of infection. Statistical differences were tested using one-way analysis of variance (ANOVA) and Kruskal–Wallis test followed by Tukey's multiple comparisons test or Dunn's multiple comparisons test, respectively. Bars represent mean, *$p < 0.05$, **$p < 0.01$, ***$p < 0.001$. Source data are provided as a Source Data file

differentiated phenotype compared to non-responding CD56$^{dim}$ NK cells, as identified by lower expression of CD57, NKG2C, and DNAM-1 and higher expression of NKp30 and NKp46 (Fig. 2d). Results from the SNE analysis were largely corroborated by results from conventional single-parameter flow cytometry data analysis (Supplementary Fig. 2a–c). Furthermore, while the expression of the anti-apoptotic molecule Bcl-2 was observed to be slightly down-regulated in responding CD56$^{bright}$ and CD56$^{dim}$ NK cells (Fig. 2b, d–f), the downregulation was modest in magnitude compared to the downregulation of Bcl-2 observed in responding T cells from the same patients (Fig. 2e, f). Additionally, responding CD56$^{bright}$ NK cells upregulated the effector molecules granzyme B and perforin (Supplementary Fig. 2d, e).

In summary, high-dimensional SNE analysis revealed responding NK cells to exhibit a less mature phenotype (i.e., predominantly CD56$^{bright}$ and less differentiated CD56$^{dim}$ NK cells) during acute DENV infection.

**The NK cell response is uncoupled from education**. Next, we investigated whether the NK cell response during acute DENV infection was coupled to KIR expression or KIR-mediated NK cell education[15,16]. Murine experimental models have suggested that uneducated NK cells dominate anti-viral responses[33]. However, our analysis revealed that NK cells responded equally well independently of the number of KIRs expressed (Supplementary Fig. 3a, b). Furthermore, their response occurred independently of KIR-mediated education (Supplementary Table 2, Fig. 3a, b). Since NKG2A can also educate NK cells[34], and our analysis revealed NKG2A expression (expressed on many immature cells) to be associated with NK cell responsiveness (Fig. 1e), we stratified for education mediated by NKG2A expression. However, also in this respect, NK cells responded independently of their educational status (Fig. 3c and Supplementary Fig. 3c). Moreover, expansions of NKG2C-expressing NK cells have previously been shown to occur in some human viral infections and are skewed towards the expression of self-KIR[6,9,35,36]. We identified two

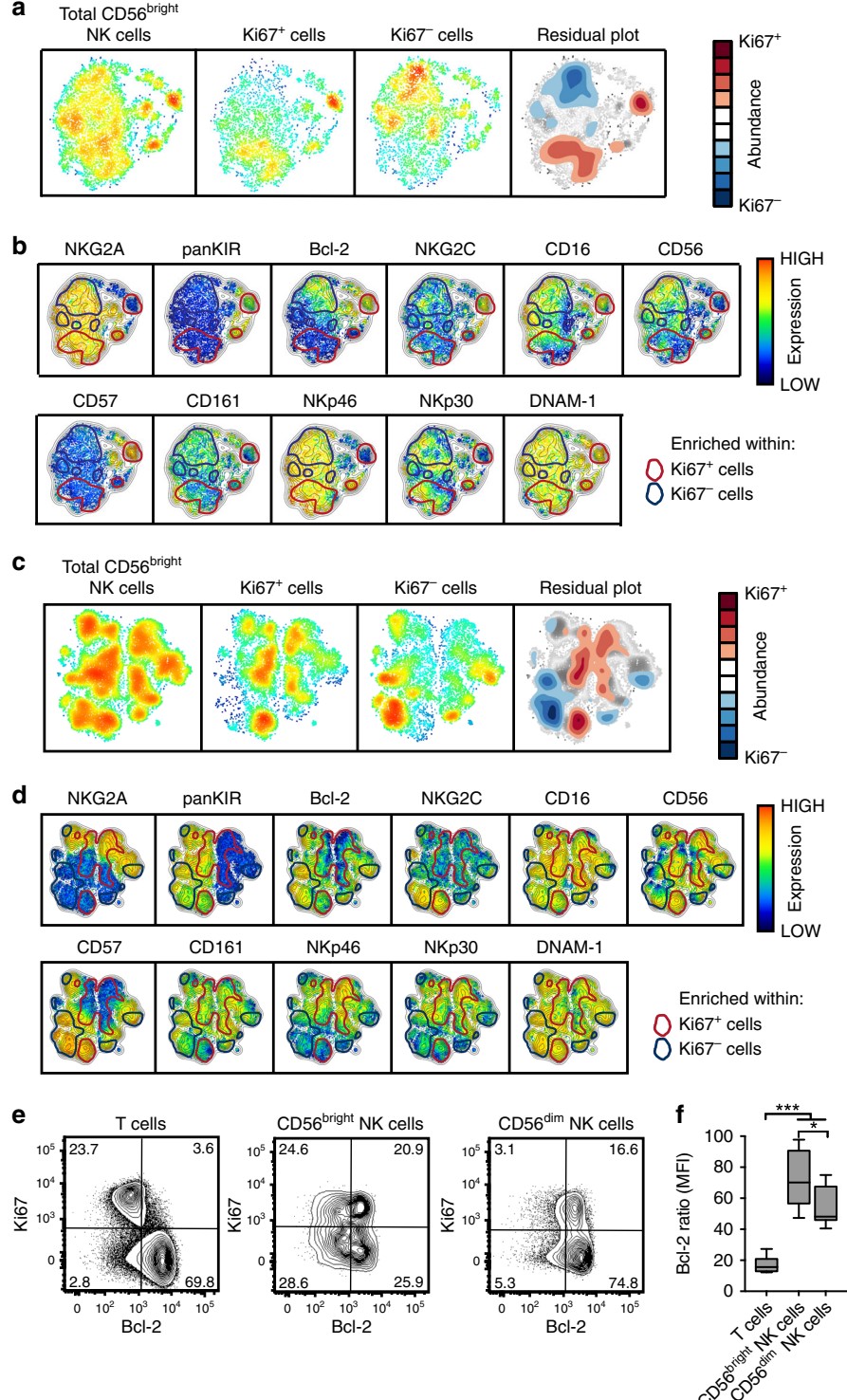

**Fig. 2** High-dimensional analysis of natural killer (NK) cells responding (Ki67+ and Ki67−) during acute dengue virus (DENV) infection. **a** Barnes–Hut stochastic neighbor embedding (Bh-SNE) maps on total CD56bright NK cells and separated for responding (Ki67+) and non-responding (Ki67−) cells. Residual plot showing the difference between responding and non-responding maps. Analysis is based on data from three representative patients at the acute phase of infection with 5000 events per subset and patient. **b** Relative expression intensities of the 11 indicated parameters that were used in the SNE analysis. Phenotypes within red circles are more common in responding NK cells and vice versa for blue circles in non-responding NK cells. **c, d** Similar analysis as in **a, b** but for CD56dim NK cells. **e** Representative flow cytometry staining for Bcl-2 and Ki67 expression of T cells, CD56bright NK cells, and CD56dim NK cells from one representative patient at the acute phase of infection. **f** Comparison of Bcl-2 expression using the median fluorescence intensity ratio (MFI, ratio of Ki67+ normalized to Ki67−) within T cells, CD56bright NK cells, and CD56dim NK cells ($n = 9$). Lowest and highest observations are displayed with the median indicated as center line. Statistical differences in **f** were tested using one-way analysis of variance (ANOVA) followed by Holm–Sidak's multiple comparison test. $*P < 0.05$ and $***p < 0.001$. Source data are provided as a Source Data file

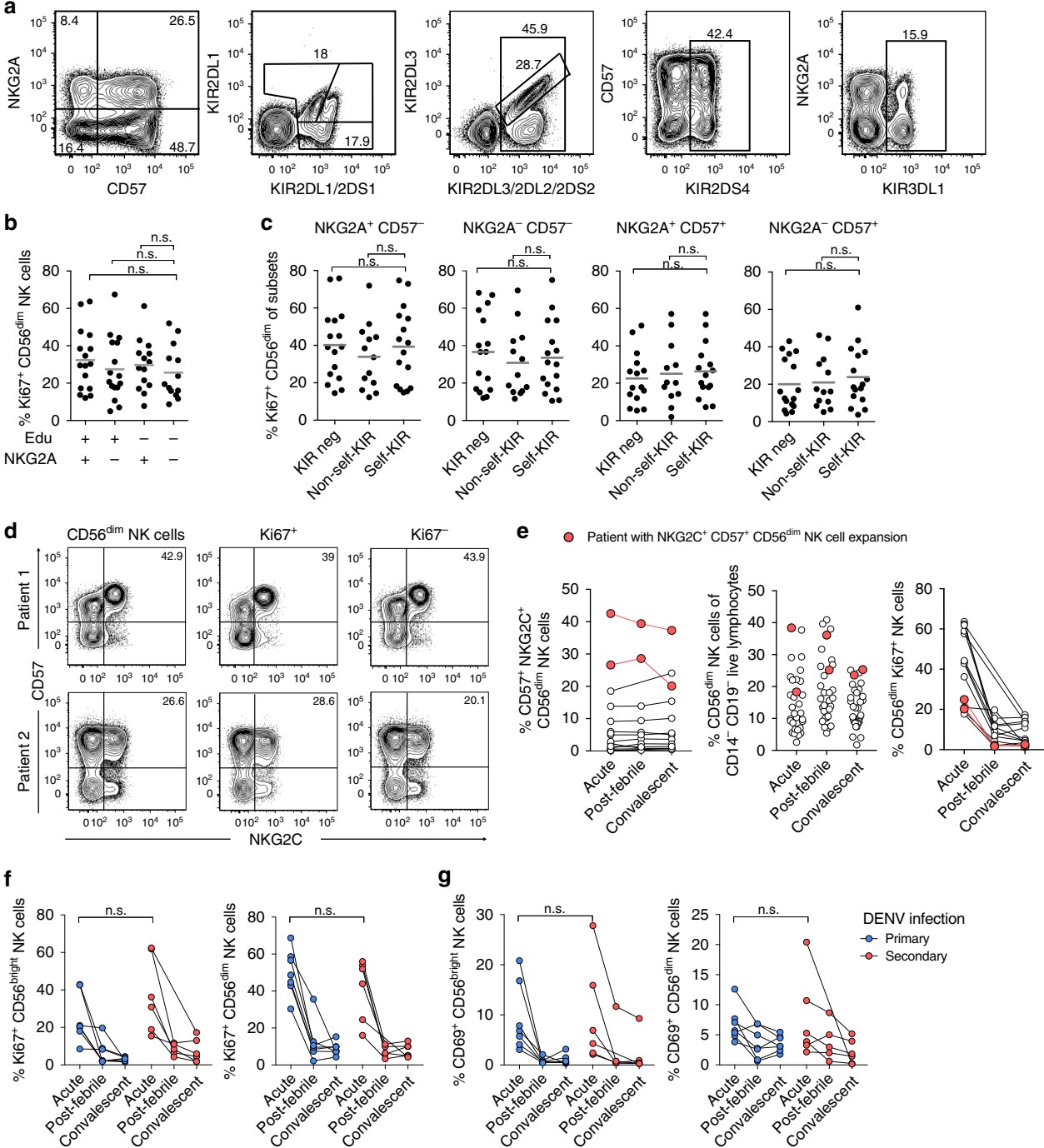

**Fig. 3** CD56$^{dim}$ natural killer (NK) cell responses during acute dengue virus (DENV) infection are uncoupled from NK cell education. **a** Gating strategy to identify NKG2A$^+$CD57$^-$, NKG2A$^-$CD57$^-$, NKG2A$^+$CD57$^+$, and NKG2A$^-$CD57$^+$ CD56$^{dim}$ NK cell subsets (first plot) expressing discrete combinations of KIR2DL1, KIR2DL1/S1, KIR2DL3, KIR2DL3/2DL2/2DS2, KIR2DS4, and KIR3DL1 (second to fifth plot). **b** Summary of data for frequency of Ki67$^+$ CD56$^{dim}$ NK cells at the acute phase of DENV infection within educated (educated via expression of self-KIR and/or NKG2A) and uneducated subsets of cells ($n = 16$). Histocompatibility leukocyte antigen (HLA) genotyping was used to assign self and non-self-KIR$^+$ subsets (see Supplementary Table 2 and Methods section). **c** Summary of data for frequency of Ki67$^+$CD56$^{dim}$ NK cells subsets expressing no KIR, one non-self-KIR (Non-self-KIR) or one self-KIR (Self-KIR) ($n = 16$). **d** Flow cytometry plots showing two donors with NKG2C$^+$CD57$^+$ adaptive-like CD56$^{dim}$ NK cell expansions within the responding (Ki67$^+$) and non-responding (Ki67$^-$) population. **e** Summary graphs highlighting the donors with adaptive-like expansions (red) as frequency of NKG2C$^+$CD57$^+$ CD56$^{dim}$ NK cells throughout infection (left), as overall CD56$^{dim}$ NK cell frequency out of lymphocytes (middle), and among responding (Ki67$^+$) cells (right). **f, g** Frequency of Ki67$^+$ and CD69$^+$ NK cells stratified according to acute primary ($n = 7$) or acute secondary ($n = 6$) DENV infection. Statistical differences were tested using one-way analysis of variance (ANOVA) followed by Tukey's multiple comparison test (**b, c**) or Mann–Whitney test (**f, g**); n. s. = not significant. Source data are provided as a Source Data file

patients with adaptive-like NK cell expansions, characterized by high expression of NKG2C and CD57 (Fig. 2d). However, these adaptive-like NK cells were not preferentially responding during acute infection in these individuals (Fig. 3d, e).

Several DENV serotypes exist and acute infection with one DENV serotype typically confers long-term protective immunity against that serotype but not against heterotypic DENV[37]. Thus, secondary (and even tertiary) acute DENV infections can occur. Since all four DENV serotypes are endemic in Singapore, this presented an opportunity to compare NK cell responses in acute primary vs. acute secondary DENV infection in patients who were classified based on rapid tests performed at the clinic at the time of patient recruitment. However, this analysis revealed the NK cell response to be similar in magnitude (Fig. 3f, g).

Taken together, the acute NK cell response to DENV infection is uncoupled from NK cell education with no evidence for an adaptive-like NK cell response to a heterotypic DENV infection.

**Acute DENV infection presents with increased IL-18 levels**. We next sought to identify mechanistic aspects of the robust NK cell activation observed in acute DENV infection. We hypothesized that the preferential activation of less mature NK cells might be driven by soluble factors. To address this, we first performed a comprehensive analysis of cytokines and soluble factors in plasma from infected patients. A vast number of factors, including pro-inflammatory and anti-viral cytokines, as well as effector molecules associated with cytotoxicity, were significantly elevated in the acute phase as compared to the convalescent phase and healthy controls (Supplementary Fig. 4). The NK cell-activating cytokines, interferon-α (IFNα), IL-12, and IL-15, were significantly elevated during the acute phase of infection (Supplementary Fig. 4), but the detected levels were relatively low. Compared to IFNα, IL-12, and IL-15, the plasma levels of IL-18 were robustly increased during the acute phase of DENV infection (Fig. 4a and Supplementary Fig. 4). This is strikingly similar to observations in a mouse model of cytomegalovirus (CMV) infection where IL-18 was critical in driving the anti-viral NK cell response[38]. Additionally, high IL-18Rα expression was observed on NK cells (Fig. 4b), especially on CD56[bright] and less mature CD56[dim] NK cell subsets during the acute phase of the infection (Fig. 4b, c). Notably, the expression levels were significantly higher than that on T cells (Fig. 4c).

We also investigated whether the skin microenvironment, one of the organs with clinical manifestation in the patients, also contained IL-18. To this end, skin blisters were induced on the forearms of patients with DENV infection, causing the detachment of epidermis from dermis and permitting minimally invasive sampling of serous fluid from the skin. Interestingly, IL-18 was found at even higher levels in skin blister fluid from DENV patients collected during the acute phase of the infection as compared to plasma (Fig. 4d). On the other hand, only low levels of IL-2, IL-12, and IL-15 could be detected (Fig. 4d). Interestingly, a recent in vitro study on murine NK cells showed a specific role for IL-18 in driving NK cell proliferation via upregulation of the nutrient transporter CD98 independently of mammalian target of rapamycin, whereas no effect on CD98 upregulation was seen for other NK cell stimulatory cytokines[39]. In line with this, we could demonstrate that NK cells from patients with acute DENV infection expressed higher levels of CD98 as compared to controls (Fig. 4e). Thus, we decided to explore further a potential role of IL-18 in driving the activation of less mature NK cells during acute DENV infection.

**IL-18 signaling is linked to the NK cell response**. In order to analyze an association between IL-18 and the activation of NK cells, we used phosphoflow to monitor signaling of key proteins and transcription factors downstream of the IL-18R known to be positive (AKTp[T308/S473], ATF2p[T69/71], NF-kBp[S529]) or negative (FOXO3Ap[S294]) regulators of the cell cycle (Supplementary Fig. 5a)[40–43]. Using peripheral blood mononuclear cells (PBMCs) from healthy controls, we identified key NK cell subsets of interest (Fig. 5a) and detected phosphorylation of the targeted proteins upon IL-18 stimulation in vitro (Fig. 5b, c). The responses observed were particularly marked within less mature CD56[dim] NK cell subsets (Fig. 5c). An enhancement of signaling downstream of IL-18 was also observed when PBMCs were exposed to acute compared to convalescent phase plasma from DENV-infected patients in combination with IL-18 (Fig. 5d, Supplementary Fig. 5c). Following this analysis, and as a critical test of our hypothesis, we determined correlates of IL-18R signaling directly on NK cells ex vivo from the DENV-infected patients. Here, we detected an IL-18 signaling signature with higher amount of phosphorylation linked to progression (pATF2, pAKT, pNF-kB) and de-repression of the cell cycle (pFOXO3A) during the acute phase of the infection (Fig. 5e, Supplementary Fig. 5d). This profile was normalized to levels of healthy controls during the post-febrile and convalescent phase (with the exception of nuclear factor-κB (NF-κB)).

Taken together, these results suggest that IL-18 contributes to the activation of less mature NK cells during acute DENV infection by coordinating activation and de-repression of the cell cycle.

**Assessment of NK cell function during acute DENV infection**. After determining the phenotype and signaling properties of NK cells during acute DENV infection, we next examined potential alterations in their functional capacity. To this end, NK cells were assessed for natural cytotoxicity responses, antibody-dependent cellular cytotoxicity, and responses following cytokine stimulation. Five NK cell functions (degranulation assessed by CD107a expression and production of IFNγ, tumor necrosis factor (TNF), macrophage inflammatory protein-1β (MIP-1β), and granulocyte–macrophage colony-stimulating factor (GM-CSF)) were simultaneously evaluated. As expected, the profile of the response changed depending on the type of stimulation and on the type of NK cell subset (CD56[bright] vs. CD56[dim]) examined (Fig. 6a). Intriguingly, NK cells retained their functional response profile during acute DENV infection (Fig. 6b). Similarly, the total frequency of responding cells as well as the multifunctional response profile was unaffected during acute infection (Fig. 6b, c). Since the NK cell response to IL-12 and IL-18 stimulation was near saturation, we also assessed the functional profile upon stimulation with lower concentrations of IL-12 and IL-18. Also in this setting, the NK cell functional response was largely retained in acute DENV infection, but with slightly lower levels of IFNγ being produced from both CD56[bright] and CD56[dim] NK cells (Supplementary Fig. 6a–c). Finally, we determined the capacity of NK cells to respond upon in vitro infection of PBMCs with DENV. Upon DENV infection, both CD56[bright] and CD56[dim] NK cells upregulated CD69, whereas only CD56[bright] NK cells showed evidence of TRAIL expression (Supplementary Fig. 7a). Furthermore, NK cells responded by producing IFNγ, TNF, and MIP-1β (Fig. 6d).

Together, these results indicate that NK cells from DENV-infected patients preserve their functional capacity throughout the infection and that NK cells have the capacity to respond to DENV infection of PBMCs in vitro.

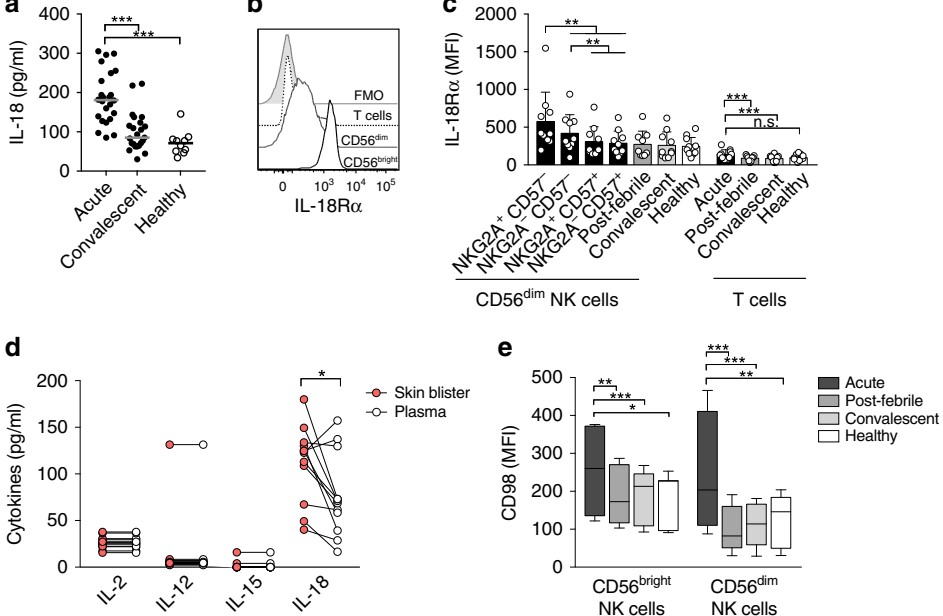

**Fig. 4** Increased levels of interleukin-18 (IL-18) during the acute phase of dengue virus (DENV) infection. **a** Summary of data showing levels of plasma IL-18 (pg/ml) during the acute and convalescent phase of DENV-infected patients ($n = 24$) compared to healthy controls ($n = 10$). **b** Representative histograms of IL-18Rα expression (mean ± SD) on CD56^bright natural killer (NK) cells, CD56^dim NK cells, and T cells during the acute phase of DENV infection. **c** Mean IL-18Rα expression (median fluorescence intensity (MFI) on CD56^dim NK cell subsets stratified for NKG2A and CD57 (black bars, left) during the acute phase of DENV infection is shown. These are compared to bulk CD56^dim NK cells from indicated phases (gray and white bars), to T cells from the same stages ($n = 10$), and to healthy controls ($n = 11$). **d** Levels of NK cell-activating cytokines from skin blister fluid collected during the acute phase of DENV infection compared to matched plasma. **e** CD98 expression (MFI) on NK cell subsets from DENV-infected patients during the acute ($n = 14$), post-febrile ($n = 12$), and convalescent phase ($n = 14$) compared to healthy controls ($n = 15$). Lowest and highest observations are displayed with the median indicated as center line. Statistical differences were tested using paired $t$ test or Wilcoxon's matched-pairs signed-rank test. *$P < 0.05$, **$p < 0.01$, and ***$p < 0.001$. Source data are provided as a Source Data file

**Specific chemokine receptor imprint on CD56^bright NK cells.** Little is known regarding NK cell homing to peripheral tissues during acute viral infections in humans. Patients with acute DENV infection often show disease manifestations in the skin, liver, and gut[44]. To investigate whether NK cells might be primed for homing to peripheral tissues during acute DENV infection, we determined the chemokine receptor profile of Ki67+/CD69+ and Ki67−/CD69− CD56^bright and CD56^dim NK cells, respectively, during the acute phase of infection. T cells were included as a positive control since upregulation of cutaneous lymphocyte-associated antigen (CLA), CCR5, and CXCR3 on DENV-specific T cells has recently been described in acute DENV infection[45], and was confirmed in the present study (Supplementary Fig. 8a, b). Strikingly, responding Ki67+ CD56^bright NK cells expressed higher levels of CLA, CCR5, CXCR6, and CCR9 as compared to both non-responding NK cells during the acute phase of infection as well as healthy controls. In contrast, CCR7 was down-modulated in responding Ki67+ CD56^bright NK cells (Fig. 7a, b). A similar pattern, albeit with lower chemokine receptor expression, was observed for responding CD56^dim NK cells. Responding CD69+ CD56^bright NK cells showed a varying pattern of chemokine receptor expression with high expression of CCR5 and CXCR6, as well as an intermediate increase in α4β7 and CCR6 (Fig. 7b). The increased chemokine receptor expression was diminished in follow-up samples where expression levels returned to those observed in healthy controls (Fig. 7b). In summary, our data revealed that responding NK cells exhibit a distinct chemokine receptor profile during acute DENV infection.

**CLA+CD69+ CD56^bright NK cells are present in skin.** The high expression of CLA suggests that responding NK cells may home to the skin during acute DENV infection. To investigate this, skin blisters were induced on the forearms of acute DENV-infected patients and healthy controls, as described above for skin blister-fluid sampling. Flow cytometry analysis of blister-fluid cells revealed that skin NK cells were enriched for NK cells of the CD56^bright subset (Fig. 8a, b). Moreover, in the patients, we found a negative correlation between the total number of NK cells per skin blister and number of days that had passed since symptom debut (Fig. 8c). Skin NK cells from patients with acute DENV infection expressed high levels of CLA and roughly half of the cells concomitantly stained positive for CD69 (Fig. 8d, f). Of note, CD69 not only identifies activated lymphocytes but also cells with a tissue-resident phenotype[46,47]. In contrast to this, fewer skin NK cells from healthy controls co-expressed CLA and CD69 (Fig. 8d, f). Finally, the chemokine receptor expression pattern on skin NK cells from patients with acute DENV infection mirrored the profile in peripheral blood with increased expression of CXCR3 and CCR5, but low expression of CCR7 and CCR10 (Fig. 8e, g). Our results suggest that responding CLA+CXCR3+CCR5+ CD56^bright NK cells have the capacity to home to the site of infection (skin) during acute DENV infection.

## Discussion

To explore the primary human NK cell response to acute DENV infection, we followed a cohort of patients from early on after onset of symptoms until disease resolution. Our results showed that NK cells responded strongly during the acute phase of infection. High-dimensional SNE analysis revealed that responding NK cells displayed a predominantly less mature phenotype (CD56^bright and immature CD56^dim cells) with retained functional capacity. Elevated levels of IL-18 in patient

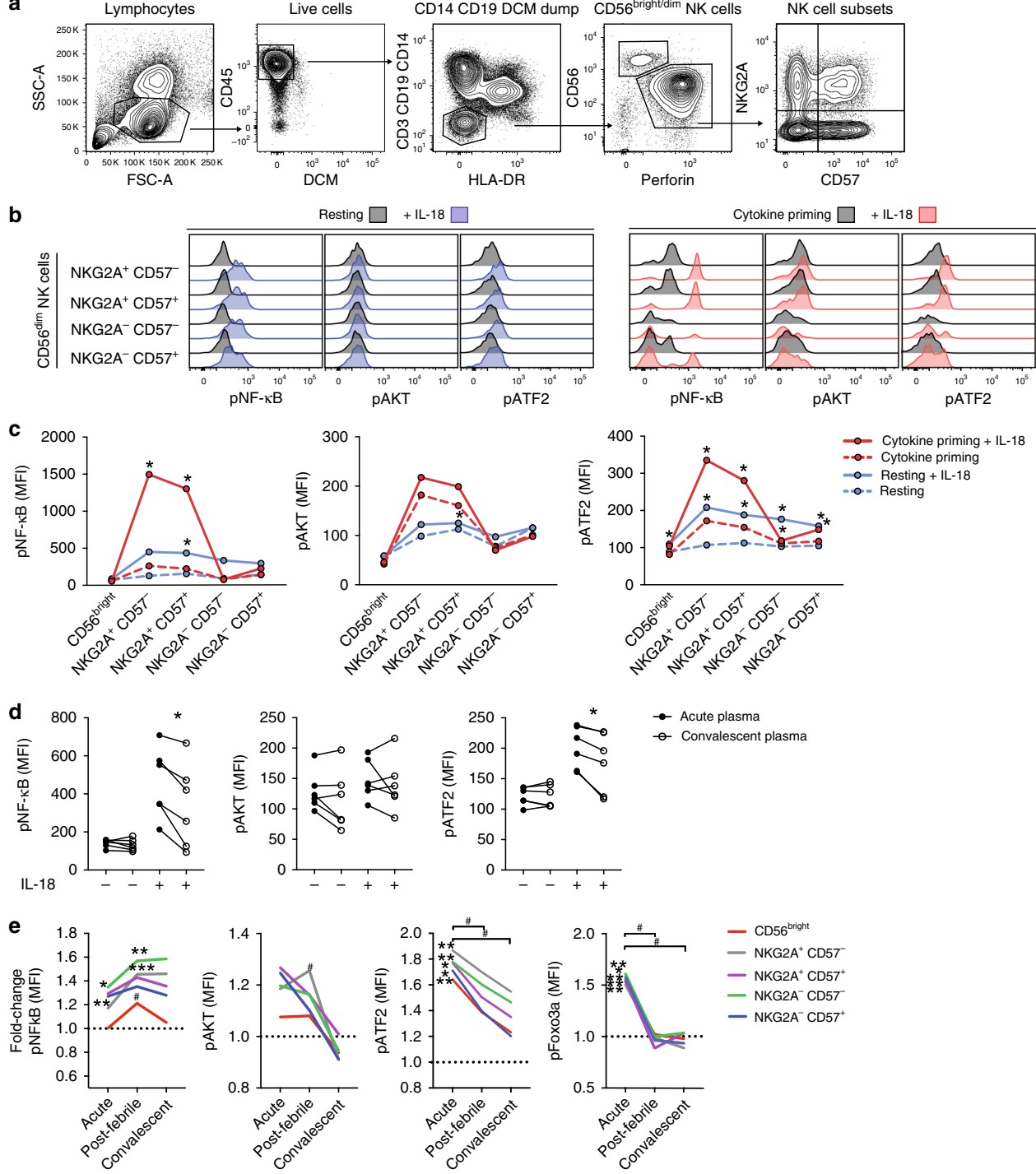

**Fig. 5** Changes in signaling downstream of the interleukin-18R (IL-18R) during acute dengue virus (DENV) infection. **a** Gating strategy to identify CD56[bright] and CD56[dim] natural killer (NK) cell subsets from total peripheral blood mononuclear cells (PBMCs) after methanol fixation. **b** Representative histograms showing staining for pNF-κB, pAKT, and pATF2 in CD56[bright] and CD56[dim] NK cell subsets from healthy controls, rested (blue) or cytokine primed, with (blue or red) or without (gray) the addition of IL-18. **c** Results summarized from experiments in **b** (n = 6). **d** Phosphoflow epitope staining of NKG2A+CD57− CD56[dim] NK cells from healthy controls that were pre-incubated with patient sera, from the acute or convalescent phase, with or without IL-18 stimulation. **e** Ex vivo expression of pNF-κB, pAKT, pATF2, and pFOXO3A in CD56[bright] and CD56[dim] NK cell subsets from the acute, post-febrile, and convalescent phase of DENV infection (n = 6). Statistical differences were tested using paired t test or Wilcoxon's matched-pairs signed-rank test. Stars (*) indicate significant differences between the non-IL-18 control compared to the IL-18-stimulated condition (**c**) or significant differences between patients and healthy controls (**e**); hashes (#) indicate significant differences between the acute phase and follow-up time points of patients with DENV infection (**e**). #P < 0.05; *p < 0.05, **p < 0.01, and ***p < 0.001. Source data are provided as a Source Data file

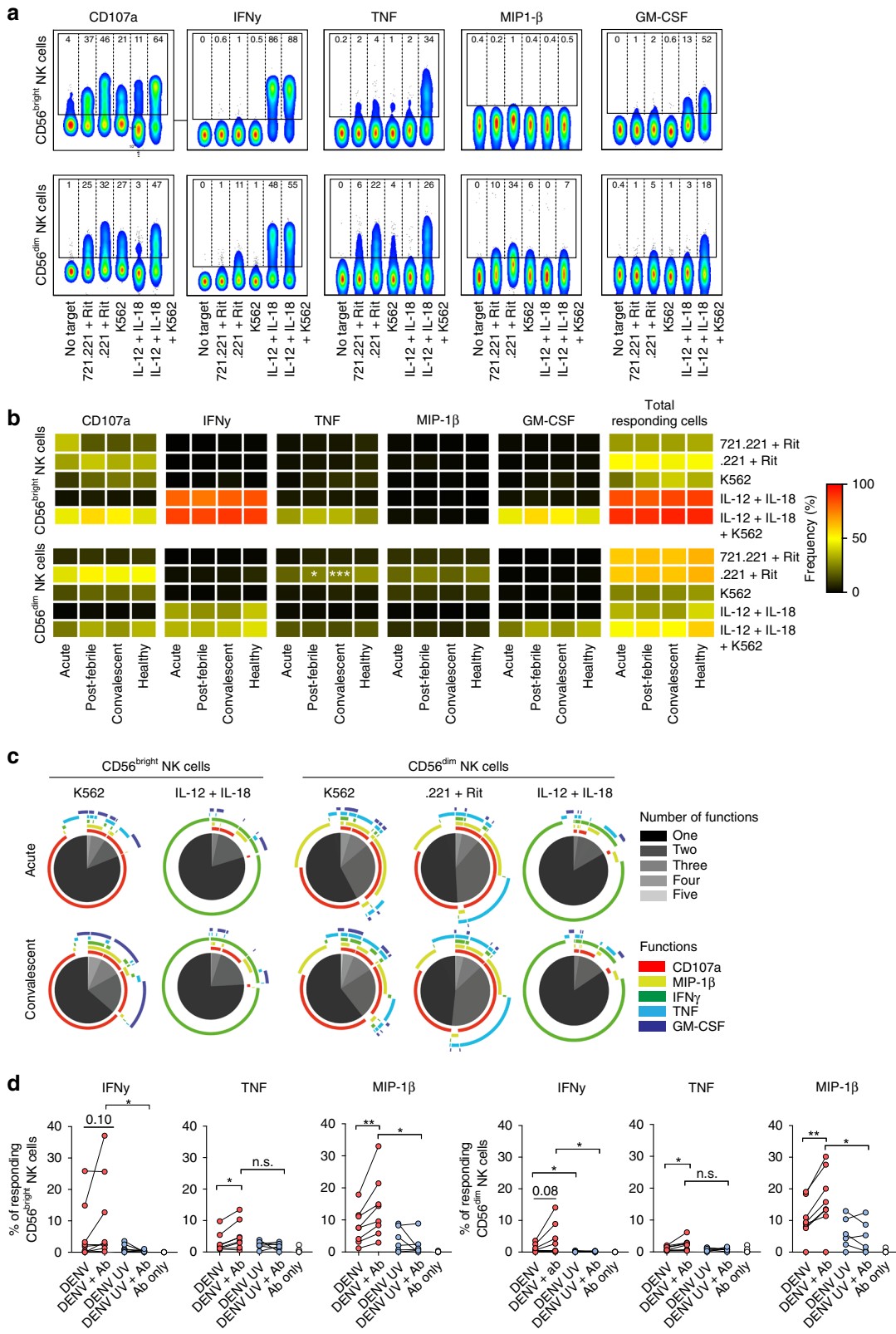

plasma and skin blister fluid, and correlates of signaling downstream the IL-18Rα in the responding NK cells, indicated an IL-18-mediated mechanism behind this specific response profile. Responding CD56[bright] NK cells furthermore presented with a chemokine receptor imprint indicative of homing to skin and liver. Concurrent with these observations, CD69[+]CLA

[+]CXCR3[+]CCR5[+] NK cells were identified in the skin during acute DENV infection.

NK cell differentiation is a continuous homeostatic process associated with phenotypic and functional changes that can adapt in response to viral infections[17,48]. Indeed, acute viral infections, such as hantavirus, chikungunya virus, human

**Fig. 6** Assessment of natural killer (NK) cell function during acute dengue virus (DENV) infection. **a** Representative flow cytometry staining shown as concatenated plots for CD107a, interferon-γ (IFNγ), tumor necrosis factor (TNF), macrophage inflammatory protein-1β (MIP-1β), and granulocyte–macrophage colony-stimulating factor (GM-CSF) for CD56$^{bright}$ and CD56$^{dim}$ NK cells after the indicated stimulations. Gates were set according to unstimulated cells (no target). Numbers within gates indicate percent positive cells. Rit, Rituximab®. **b** Heat map summarizing the frequency of CD56$^{bright}$ NK cells and CD56$^{dim}$ NK cells responding to the indicated stimulations (K562 cells, $n = 4$; 721.221 cells, $n = 9$; 721.221 cells + Rituximab®, $n = 9$; IL-12 + IL-18, $n = 4$; IL-12 + IL-18 + K562 cells, $n = 4$) during the acute, post-febrile, and convalescent phase of DENV infection, and healthy controls. **c** Pie charts showing the number of functions, as defined by Boolean gating, simultaneously exhibited by CD56$^{bright}$ (left panels) and CD56$^{dim}$ (right panels) NK cells either at the acute (upper panels) or convalescent phase (lower panels) of DENV infection in response to the indicated stimuli. The number of functions (one to five) is shown in different shades of gray, and the outer arcs indicate which functions were detected within each pie. **d** Frequency of CD56$^{bright}$ NK cells and CD56$^{dim}$ NK cells responding with IFNγ, TNF, and MIP-1β production upon in vitro infection of healthy donor peripheral blood mononuclear cells (PBMCs) with live DENV (DENV), DENV pre-incubated with the chimeric 4G2 monoclonal antibody (mAb) (DENV + Ab), ultraviolet (UV)-inactivated DENV (DENV UV), UV-inactivated DENV pre-incubated with the chimeric 4G2 mAb (DENV UV + Ab), and with 4G2 antibody only (Ab) for 24 h. Medium control was subtracted. Statistical differences were tested using paired $t$ test or Wilcoxon's matched-pairs signed-rank test. *$P < 0.05$ and **$p < 0.01$; n.s. not significant. Source data are provided as a Source Data file

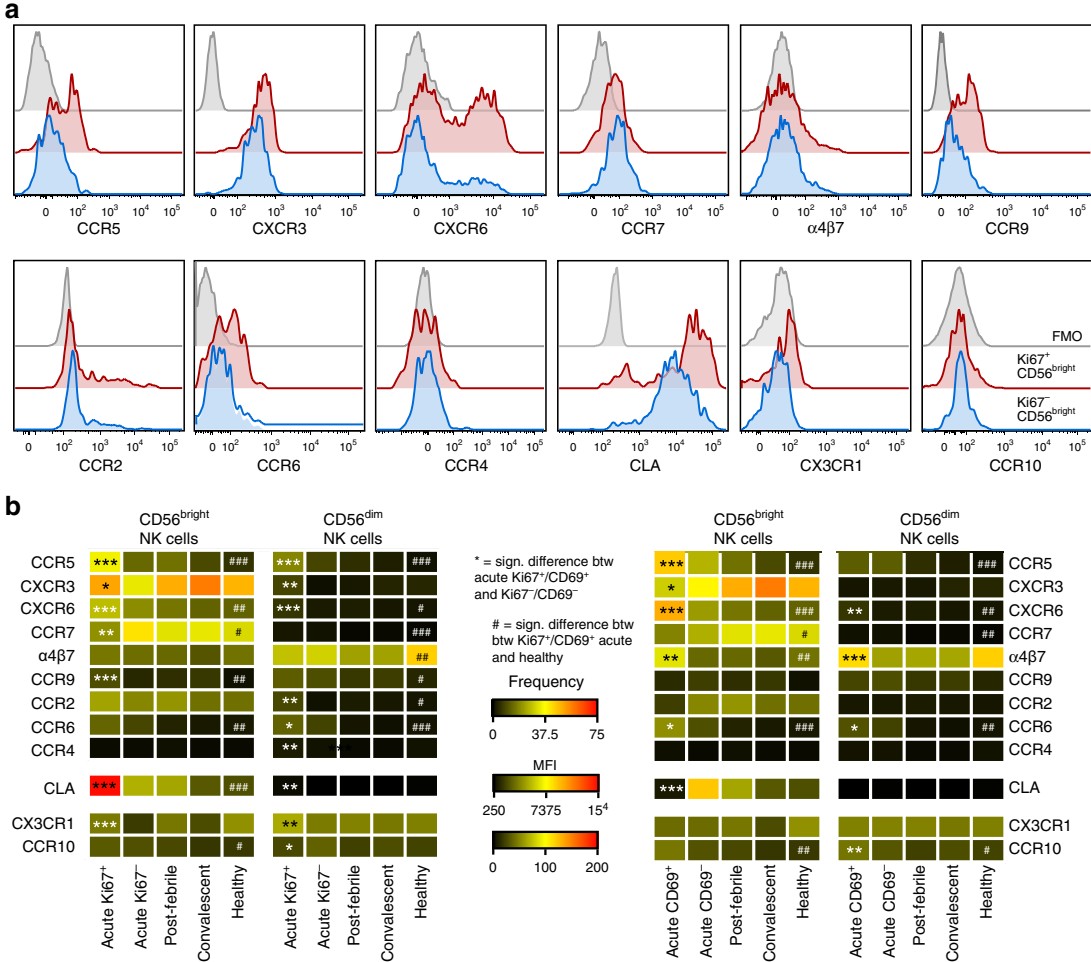

**Fig. 7** Distinct chemokine receptor imprint of CD56$^{bright}$ natural killer (NK) cells during acute dengue virus (DENV) infection. **a** Representative histograms showing chemokine receptor staining on responding (Ki67$^+$, red) and non-responding (Ki67$^-$, blue) CD56$^{bright}$ NK cells and the respective fluorescence minus one (FMO) controls (gray). **b** Heat map summarizing (median) the expression of each of the 12 chemokine receptors on CD56$^{bright}$ (left) and CD56$^{dim}$ (right) NK cells. This is subdivided into Ki67$^{+/-}$ (two left panels) and CD69$^{+/-}$ (two right panels) during the acute phase compared to the post-febrile and convalescent phase of DENV infection ($n = 10-21$) and healthy controls ($n = 12-16$). Frequencies are shown for all chemokine receptors except CLA, CX3CR1, and CCR10. Statistical differences were tested using paired $t$ test or Wilcoxon's matched-pairs signed-rank test and unpaired $t$ test or Mann–Whitney test. Stars (*) represents Ki67$^+$ and CD69$^+$ compared to Ki67$^-$ and CD69$^-$, respectively. *$P < 0.05$, **$p < 0.01$, and ***$p < 0.001$. Hashes (#) represent Ki67$^+$ or CD69$^+$ compared to healthy controls, #$p < 0.05$, ##$p < 0.01$, ###$p < 0.001$. Source data are provided as a Source Data file

immunodeficiency virus type 1, and Epstein–Barr virus (EBV) infection, can lead to expansion of adaptive-like NK cell populations exhibiting a mature differentiation phenotype[6,9,10,12,49]. Such adaptations have often been attributed to underlying CMV infection where responding mature cells are characterized by high levels of NKG2C, CD57, and DNAM-1, and typically do not express NKG2A and natural cytotoxicity receptors[10,36,50]. In our analysis, no expansions of adaptive-like NK cells were observed in acute DENV infection. Corroborating these findings, two patients identified with

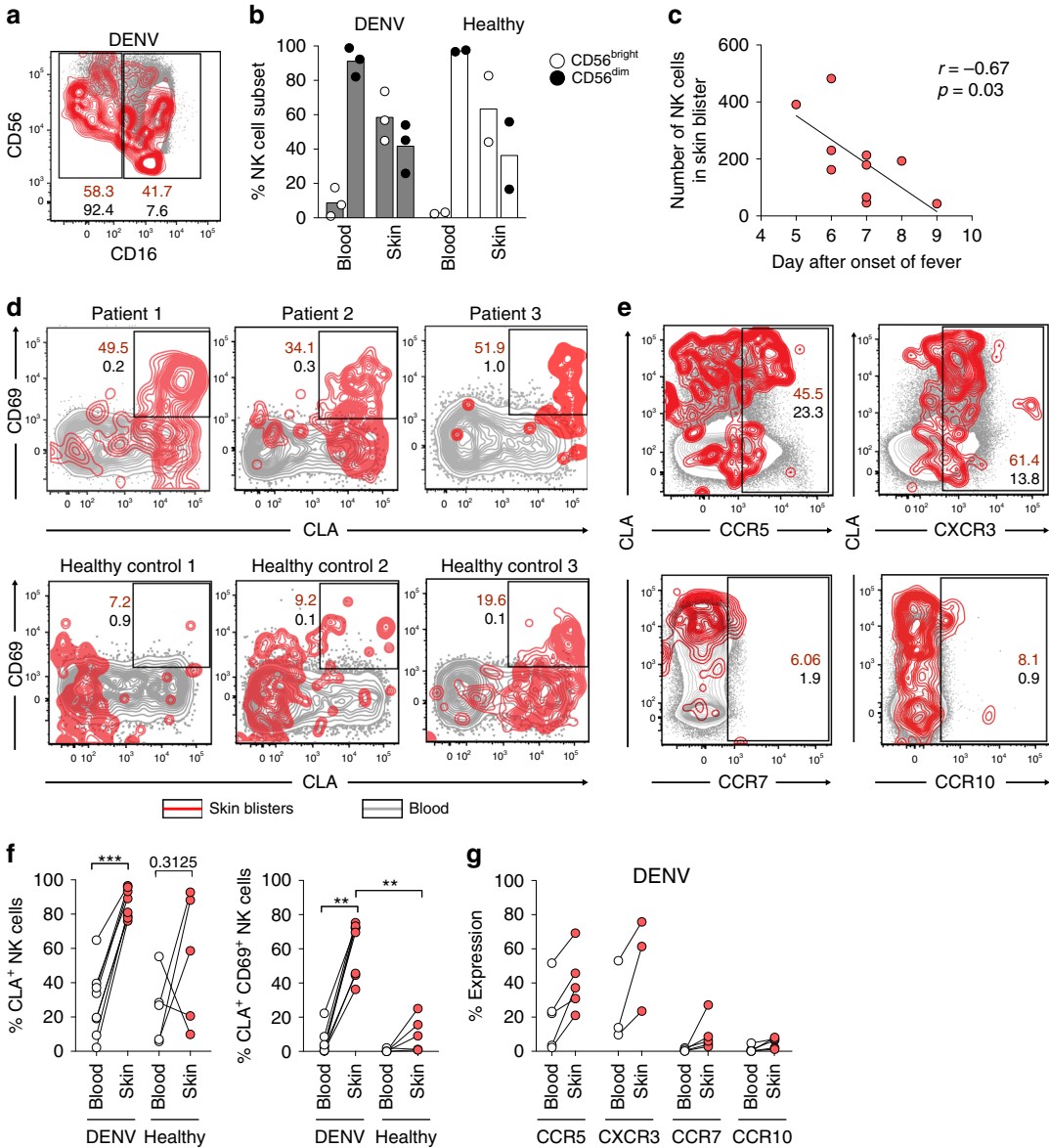

**Fig. 8** CLA+CXCR3+ CCR5+ CD69+ natural killer (NK) cells are present in the skin during acute dengue virus (DENV) infection. **a** Representative plot showing CD56bright NK cells and CD56dim NK cells from isolated from skin blisters (red overlay) and matched peripheral blood (gray background) that is summarized in **b** (DENV patients n = 3, healthy controls n = 2). **c** Spearman's correlation between number of NK cells isolated from skin blisters and the day of sample collection (day after symptom debut). **d** Representative staining for CD69 and CLA (upper panels) on NK cells from skin blister fluid (red overlay) and peripheral blood mononuclear cell (PBMC) (gray background) of three patients with acute DENV infection, and three healthy controls (lower panel). **e** Representative staining for CCR5, CXCR3 (upper panel), CCR7, and CCR10 (lower panel). **f** Summary data of d for CLA and CD69 expression on NK cells from DENV patients (n = 8) and healthy controls (n = 5). **g** Summary data of e for chemokine receptor expression on NK cells from DENV-infected patients (n = 3–5). Statistical differences were tested in using paired t test, Wilcoxon's matched-pairs signed-rank test and Mann–Whitney test. **P < 0.01 and ***p < 0.001. Source data are provided as a Source Data file

adaptive-like NK cell expansions demonstrated no preferential responsiveness attributed to these expansions during acute infection. Moreover, we did not find the NK cell response to be superior during acute secondary DENV infection, which would further argue against an adaptive-like NK cell component in the response to acute DENV infection. Rather, a high-dimensional 11-parameter SNE analysis revealed responding NK cells to exhibit a less differentiated phenotype with high expression of NKG2A and NCRs, and low expression of CD57, NKG2C, and DNAM-1. As such, our results are more in line with the NK cell response following live attenuated yellow fever virus (YFV) vaccination or that seen in acute EBV infection causing infectious mononucleosis[7,51].

NK cells expressing inhibitory KIRs or NKG2A are known to undergo an education process leading to increased functionality[15,16]. However, it remains unclear if NK cell education is a determining factor for NK cell responses in infectious or inflammatory conditions in humans. In mice infected with *Listeria monocytogenes*, educated and uneducated NK cells produced equivalent amounts of IFNγ[52], and primarily uneducated NK cells controlled murine CMV infection[33]. Human NK cells are predominantly educated via NKG2A and inhibitory KIRs[16]. Our data show that educated and uneducated NK cells responded to a similar extent during acute DENV infection. This is in contrast to the adaptive-like NK cell expansions appearing in certain infections. The latter are often characterized by the absence of NKG2A

and the expression of self-specific inhibitory (educating) KIRs[6,50]. Instead, our results corroborate recent reports studying NK cell responses following YFV vaccination and tick-borne encephalitis virus (TBEV) infection[7,8]. In this regard, it is known that less differentiated NK cells are more responsive to cytokine stimulation as compared to more differentiated cells and that NK cell education status does not impact cytokine responsiveness[17]. Since the observed response occurred independently of education, it is plausible that it is primarily driven by soluble factors. Indeed, we could detect significantly elevated levels of several cytokines known to promote NK cell activation during the acute phase of infection, including IL-18.

In respect to the elevated levels of IL-18 in plasma and skin, we hypothesized that IL-18 might drive NK cell activation and proliferation of less mature NK cell subsets. This hypothesis was further underscored by the observed upregulated levels of the nutrient transporter CD98 on NK cells during acute DENV infection. IL-18 was recently shown to drive murine NK cell proliferation via upregulation of CD98[39]. Therefore, we explored evidence of IL-18 signaling by studying NF-κB, AKT, ATF2, and FOXO3A phosphorylation, signaling molecules that have all been described to be involved in cell cycle regulation and survival[40–43]. Indeed, less mature NK cells responded via these pathways upon IL-18 stimulation in vitro. Furthermore, we found that NK cells analyzed ex vivo during the acute phase of infection had significantly elevated levels of phosphorylated ATF2 and FOXO3A. ATF2 induces proteins promoting the cell cycle, such as cyclins and the anti-apoptotic molecule Bcl-2[41]. FOXO3A, located downstream of AKT signaling, actively represses the cell cycle in the nucleus by inhibition of several proteins (cyclins, cyclin-dependent kinases) and is inactivated upon phosphorylation via cytoplasmic translocation followed by degradation[43]. NF-κB, also found to be elevated in the patients, is known to trigger the translation of adhesion and anti-apoptotic proteins, inflammatory cytokines, certain chemokine receptors (CCR5), and activation markers (CD69)[42]. These were all found to be elevated on NK cells during the acute phase of infection, thereby promoting survival and migration. Notably, the NK cell-activating cytokines, IFNα, IL-12, and IL-15, were also elevated during the acute phase of the infection. Even though we could only detect low levels of these cytokines, they may also take part in promoting NK cell proliferation. For example, cytokine-activated NK cells upregulate the IL-2Rα subunit and respond to low concentrations of IL-2[53]. Moreover, signaling cascades are complex and also other stimuli may induce phosphorylation of the investigated signaling molecules. Nevertheless, the results suggest that IL-18 contributes towards driving the detected NK cell response during acute DENV infection.

NK cell homing to peripheral tissues plays a critical role in host defense against many pathogens in murine models[24–27,29,54,55]. This trafficking is orchestrated by unique sets of chemokines being produced by specialized cell types in different organs[56]. NK cell homing patterns during acute infections in humans are not well described. Nevertheless, in chronic inflammatory and autoimmune conditions in humans it has been suggested that primarily CD56[bright] NK cells use CXCR3, CCR5, and CCR8 to localize inflamed skin and joints[57–59]. In light of this, we undertook a comprehensive assessment of chemokine receptor expression on NK cells during acute DENV infection with the aim to identify chemokine receptor expression for tissues that are affected during acute DENV infection. The results showed that responding CD56[bright] NK cells transiently upregulated CLA, CCR5, CXCR6, and CCR9. Intriguingly, this imprint was less pronounced for CD56[dim] NK cells. The pattern of chemokine receptor expression on CD56[bright] NK cells mirrored the pattern displayed by responding T cells during acute DENV infection, as

previously reported[45] and confirmed in the present study. Since priming of naive T cells occurs in lymph nodes during viral infections, and since CD56[bright] NK cells are enriched at such locations[60], it is tempting to speculate that the priming of CD56[bright] NK cells could take place there. If so, this would provide a plausible explanation for the differences in chemokine receptor profiles observed for CD56[bright] NK cells when compared to CD56[dim] NK cells. Interestingly, we could also show that primarily CD56[bright] NK cells were present in the skin during acute infection. Furthermore, NK cells appeared to be rapidly recruited to skin during acute infection since a negative correlation was present between NK cell numbers per skin blister and days after symptom debut. Finally, the NK cells present in skin blister fluid expressed high levels of CLA, CXCR3, and CCR5.

Previous studies of NK cells in acute DENV infection demonstrated increased levels of CD69 expression on NK cells[61,62]. Our results are in line with those earlier studies. However, our data suggest expression of Ki67 to be a more robust indicator of responding NK cells, since a larger fraction of both CD56[bright] and CD56[dim] NK cells upregulated Ki67 early after symptom debut. As previously shown after YFV vaccination, Ki67 and CD69 were expressed in a largely mutually exclusive pattern, suggesting these subsets of NK cells to represent different phases of the response[7]. CD69 is an early activation marker for cells in peripheral blood, whereas Ki67 is upregulated at later time point when cells enter the cell cycle. During acute DENV infection, the detected Ki67 response was several orders of magnitude stronger compared to the recently reported NK cell responses in other flavivirus infections, including YFV (attenuated YFV 17D) and TBE infection[7,8], and more in line with the potent responses observed in, for example, acute hantavirus infection[6]. This might be a reflection of how clinically severe the distinct acute phases of these respective infection are.

Given the strong proliferative response observed during acute DENV infection, it may be considered surprising that the frequency of NK cells remains constant in peripheral blood. Furthermore, NK cells retained expression of the anti-apoptotic molecule Bcl-2 to a higher degree compared to responding T cells. This may suggest active homing of responding cells to peripheral tissues as an explanation for the constant levels of NK cells in peripheral blood. Indeed, we could show that responding NK cells, which share phenotypic features with their peripheral blood counterparts, were present in the skin during acute DENV infection. Notably, NK cells from skin blister fluid expressed high levels of CD69 during the acute phase of the infection, a marker that has recently been associated with lymphocyte tissue residency[46,47]. Interestingly, besides skin-resident cells such as macrophages, Langerhans cells, and dendritic cells as potential source of IL-18, keratinocytes can also produce IL-18. Furthermore, and in response to IFN-γ and TNF, keratinocytes can produce ligands for CXCR3 (CXCL9, 10, and 11) and CCR5 (CCL3, 4, and 5)[19,63,64]. This may further augment the recruitment of CXCR3- and CCR5-expressing cells, such as NK cells and T cells. Previously cited literature in other conditions than acute viral infections has indicated a role for CCR5 in skin homing. In line with this, we detected increased expression of CCR5 on responding CD56[bright] NK cells, suggesting that not only CLA but perhaps also CXCR3 and CCR5 promote NK cell homing to skin during acute DENV infection, receptors that have recently been shown to be also expressed by DENV-specific T cells homing to the skin[45]. A recent report also showed a role for CXCR6 on NK cells in homing to skin after varicella zoster virus (VZV) skin challenge[65], whereas we could show that responding circulating NK cells upregulated CXCR6. Unfortunately, we did not have the possibility to investigate CXCR6 on NK cells in skin. However, since CXCR6 has primarily been studied in relation to liver

homing, it is plausible that CXCR6-expressing NK cells also infiltrate the liver during DENV infection. Especially since many patients display elevated levels of liver enzymes during acute DENV infection. Previous literature has suggested a role for CCR8 in NK cell homing to skin; however, because of technical difficulties in performing flow cytometry staining for this receptor, we did not manage to assess a possible role for CCR8.

Intriguingly, an early report has associated lower levels of NK cells in peripheral blood with dengue hemorrhagic fever[62], a more severe clinical presentation of acute infection than DF. Thus, it is tempting to speculate that NK cells in the former patients have entered peripheral tissues and contribute to clinical disease. In the present study, patients with DF were included. Hence, no correlations with disease activity (i.e., DF vs. dengue hemorrhagic fever) could be performed. Nevertheless, our results provide as a platform for future pathogenesis-oriented studies comparing larger groups of patients, also with different disease presentations. Such future studies should focus on determining if cytokine-mediated or cytotoxic (such as rejection of infected cells) NK cell responses are most important in disease pathogenesis. In this regard, it is interesting to note that NK cells infiltrating skin after VZV antigen challenge were CD107a$^+$, suggestive of an ongoing cytotoxic response[65].

In summary, CD56$^{bright}$ and less mature CD56$^{dim}$ NK cells were the main responding NK cells during acute DENV infection in humans. The response correlated with increased levels of IL-18 in plasma and skin blister fluid during acute DENV infection and signaling downstream of IL-18Rα. A specific imprint of peripheral tissue homing was uniquely present within the CD56$^{bright}$ NK cell compartment. Concomitantly, a corresponding NK cell subset was identified in skin of patients during the acute phase of infection. These results represent previously uncharacterized features of NK cells during an acute viral infection in humans.

## Methods

**Study approval.** Peripheral blood samples were longitudinally collected at Tan Tock Seng Hospital from 32 patients with DENV infection (DF and one patient with dengue hemorrhagic fever) throughout the acute, post-febrile, and convalescent phases of infection. Twenty-six community-matched healthy controls were also sampled for peripheral blood and included in the study. Skin blister samples were collected from eight DENV-infected patients during the acute phase of infection as well as corresponding samples from five healthy controls. Ethical regulations needed for working with human participants were obtained. All study subjects gave written informed consent. The experiments were conducted according to the Declaration of Helsinki and approved by the Singapore National Healthcare Group ethical review board (DSRB 2013/00209 and DSRB 2008/00293). Detailed clinical information of the DENV-infected patients is presented in Supplementary Table 1.

**Dengue diagnosis.** The patients were diagnosed with acute DENV infection through routine clinical diagnostics (detection of DENV RNA by reverse transcription-polymerase chain reaction or NS1 antigen by enzyme-linked immunosorbent assay (ELISA)). Patients that fulfilled the World Health Organization clinical criteria for acute DENV infection and were positive for immunoglobulin M (IgM) and IgG serology (Panbio Dengue Duo Casette) were also included in the study. Secondary DENV infection was determined based on results from the Panbio Dengue Duo IgM and IgG Rapid Strip Test or the SD Bioline NS1 antigen + antibody combo (IgM and IgG antibody combination) test.

**PBMC isolation.** Blood was collected in EDTA-treated vacuum tubes at three different time points (acute, post-febrile, and convalescent) after the onset of fever. PBMCs were isolated using Ficoll-Hypaque gradient centrifugation and cryopreserved in fetal calf serum (FCS) (Thermo Fisher Scientific) with 10% dimethyl sulfoxide (Thermo Fisher Scientific) in liquid nitrogen for later analysis.

**Skin blister induction and cell isolation.** Suction skin blisters were induced on the forearm of DENV-infected patients using skin suction chambers (Medical Engineering, Royal Free Hospital) and a clinical suction pump[66]. Briefly, a negative pressure of 25–40 kPA was applied to the skin for 2 to 4 h until a unilocular blister was formed. The blister was covered overnight with a rigid adhesive dressing. After 18 to 24 h, the accumulated fluid inside the blister was aspirated and cells pelleted.

After removal of the supernatant, the pellet was resuspended in the AIM-V medium (Thermo Fisher Scientific) supplemented with 2% AB human serum, and the cells were analyzed by flow cytometry as described below.

**Measurement of plasma proteins during acute DENV infection.** Plasma concentrations of IL-1β, IL-2, IL-6, IL-10, IL-12, IL-15, IL-18, TNF, IFNγ, granzyme A, granzyme B, C5a/C5, vascular endothelial growth factor, and B cell-activating factor were assessed using a custom-made magnetic luminex screening assay (R&D Systems) according to the manufacturer's instruction. IFNα was measured with a multi-subtype type I interferon ELISA kit (PBL Assay Science) and IL-21 by using an IL-21 ELISA kit (AH Diagnostics), both according to the protocols provided by the manufacturers.

**KIR and KIR ligand genotyping.** Genomic DNA was isolated using a DNeasy Blood & Tissue kit (Qiagen). KIR genotyping and KIR ligand determination were performed using PCR-SSP (single-specific primer) technology with KIR and KIR HLA ligand kits (Olerup) according to the manufacturer's protocol. Alternatively, *KIR* genotyping was performed using the PCR-SSO (sequence-specific oligonucleotide) luminex-based method (OneLambda, Thermo Fisher). The KIR and HLA genotypes of the patients are listed in Supplementary Table 2.

**Flow cytometry.** Ex vivo isolated PBMCs were thawed and stained with fluorescently labeled antibodies. See Supplementary Table 3 for a complete list of antibodies used. Biotinylated and purified antibodies were visualized using streptavidin-coupled or anti-IgM secondary antibodies, respectively. Fixable LIVE/DEAD Aqua or Blue dead cell stain kits (Life Technologies) were used to exclude dead cells. For extracellular staining, samples were incubated for 20 min at room temperature or for chemokine receptor staining for 30 min at 4 °C or 37 °C. After fixation/permeabilization using fixation/permeabilization buffer (eBioscience), PBMCs were stained intracellularly for 30 min in FACS Permwash buffer (eBioscience) using the antibodies listed for intracellular staining in Supplementary Table 3. The following reagent was obtained through the NIH AIDS Reagent Program, Division of AIDS, NIAID, NIH: anti-human α4-β7 integrin monoclonal (Act-1) (cat#11718) from Dr. A.A. Ansari[67]. Samples were acquired on BD LSR Fortessa equipped with five lasers (BD Biosciences).

**Functional analysis.** Cryopreserved PBMCs were thawed in complete RPMI medium, meaning RPMI-1640 medium (Thermo Fisher Scientific) supplemented with 10% FCS (Thermo Fisher Scientific) and 1 mM L-glutamine (Invitrogen). PBMCs were either rested or stimulated overnight with IL-12 (PeproTech) and IL-18 (R&D Systems) at 37 °C and 5% CO$_2$. For results from functional experiments shown in Fig. 6, IL-12 was used at 10 ng/ml and IL-18 at 100 ng/ml. For results from functional experiments shown in Supplementary Fig. 6, concentrations used are indicated in the figure. After overnight incubation, 10$^5$ target cells, either K562 cells or 721.221 (.221) cells (both from ATCC), with or without Rituximab® (Rit, 1 μg/ml), were added to 10$^6$ rested or cytokine-stimulated PBMCs for additional 6 h. Anti-CD107a FITC (BD Bioscience) was present throughout the assay. Monensin and brefeldin A (BD Biosciences) were added during the final 5 h. PBMCs were subsequently stained with additional antibodies and analyzed by flow cytometry as described above.

**Propagation of DENV stock.** C6/36 mosquito cells were grown using supplemented Leibovitz's L-15 medium (5% FCS, 1% PeSt, and 2% tryptose phosphate (all from Thermo Fisher Scientific)) and infected with DENV type 2 (strain 4397-11). Infected cells were incubated for 1 week. Supernatants were harvested from infected and uninfected mosquito cells and stored at −80 °C.

**Infection of PBMC with DENV.** PBMCs from healthy donors were isolated by density centrifugation (Ficoll-Hypaque from GE Healthcare). DENV stock was exposed to ultraviolet (UV) light for 30 s in order to obtain an inactivated DENV control. Supernatants from uninfected mosquito cells were used as mock infection for uninfected controls (medium control). Viruses were diluted in RPMI medium (RPMI-1640 medium (Thermo Fisher Scientific) supplemented with 10% FCS (Thermo Fisher Scientific), 1% PeSt (Thermo Fisher Scientific) and 1 mM L-glutamine (Invitrogen)) with or without the infection enhancing chimeric 4G2 monoclonal antibody (0.38 μg/ml) and incubated for 30 min at 4 °C. After the incubation, PBMCs were pelleted and resuspended in the medium containing DENV, UV-treated DENV, or mock, with or without the 4G2 monoclonal antibody. The cells were then incubated for 2 h at 37 °C and 5% CO$_2$. Subsequently, cells were centrifuged and washed once with complete RPMI medium. The PBMCs were plated in duplicates as 10$^6$ cells per well and incubated overnight at 37 °C and 5% CO$_2$ in complete RPMI medium. PBMCs were subsequently stained with antibodies and analyzed by flow cytometry as described above.

**Detection of phospho-epitopes for cell signaling analysis.** Ex vivo isolated PBMCs from healthy donors were thawed and rested in complete RPMI medium for 1 h at 37 °C and 5% CO$_2$. Subsequently, PBMCs were stimulated with IL-18 (100 ng/ml; R&D Systems) or PBS (non-stimulated control) for 15 min at 37 °C

and 5% $CO_2$ in the presence of LIVE/DEAD Yellow (Life Technologies). For in vitro experiments mimicking acute in vivo priming, NK cells were first exposed to IL-2 (Peprotech), IL-12 (Peprotech), IL-15 (Peprotech), and IFNα (PBL) at 2 ng/ml or a pool of patient plasma diluted 1:2 in RPMI medium for 1 h. For ex vivo analysis, PBMCs were thawed and directly fixed without resting. IL-18 stimulation was stopped with formaldehyde 2% (Polysciences) and cells were stained for extracellular markers. After cell permeabilization with methanol, staining for phospho-epitopes and perforin was performed. Fc receptor blocking (Miltenyi) was included in all stains.

**Flow cytometry data analysis.** Flow cytometry data analysis was performed using FlowJo version 9.9.4 (TreeStar). Post-processing was performed using SPICE version 5.3 (provided by M. Roederer and J. Nozzi, NIAID, NIH) as well as R version 3.3.1 (The R Foundation of Statistical Computing). SNE analysis, a nonlinear dimensionality reduction method making two-dimensional representations at the single-cell level of highly complex data, was used to identify multivariate relationships as previously described[32].

**Statistics.** Statistical analysis was performed using GraphPad Prism version 7 (GraphPad Software). Data were first probed for normality using the D'Agostino–Pearson omnibus test. For normally distributed data, paired or non-paired, the Student's t test or when comparing several groups one-way analysis of variance followed by the Holm–Sidak's multiple comparison test were used. For non-normally distributed data, the Wilcoxon's matched-pairs signed-rank test was used for matched pairs of data and the Mann–Whitney test for unmatched pairs of data. For multiple comparisons, Kruskal–Wallis test followed by Dunn's multiple comparisons test were used. Correlation analysis of non-normally distributed data was performed with the Spearman's correlation, and statistical significance exhibited $*P < 0.05$, $**P < 0.01$, and $***P < 0.001$.

**Reporting summary.** Further information on research design is available in the Nature Research Reporting Summary linked to this article.

## Data availability

The authors declare that the data supporting the findings of this study are available in the article, the Supplementary information files, or upon request to the authors. The source data underlying Figs. 1b–f, 2f, 3b, c, e–g, 4a, c–e, 5c–e, 6b, d, 7b, 8b, c, f, g and Supplementary Figs. 1b–e, 2b, c, e, 3a, b, 4a, 5c, 6a–c, 7a, 8b are provided as a Source Data file.

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

## Acknowledgements

We thank Dr. Marie Schaffer for help with performing KIR and KIR ligand typing. This work was funded by the Swedish Research Council, the Swedish Cancer Society, the Swedish Foundation for Strategic Research, the Swedish Society for Medical Research, the Cancer Research Foundations of Radiumhemmet, Knut and Alice Wallenberg Foundation, the Novo Nordisk Foundation, the Center for Innovative Medicine at Karolinska Institutet, the Stockholm County Council, Karolinska Institutet, and by an Open Fund-Individual Research Grant (OF-IRG R-913-301-439-213) of the Singapore National Medical Research Council awarded to L.R. and Duke-NUS Medical School, Singapore. Open access funding provided by Karolinska Institute.

## Author contributions

All authors contributed to study concept, set up, and design, as well as interpretation of data. C.L.Z., M.C., and C.S.-R. planned experiments, and acquired and analyzed data. K.-W.C., M.A.I., L.M.Q., N.M., Y.-S.L., D.C.L., J.K., P.A.M., and H.-G.L. acquired data. C.L. Z. and N.K.B. drafted the manuscript. L.R. and N.K.B. supervised the study. All authors contributed to critical revisions and approved the final version of the manuscript.

## Additional information

**Competing interests:** The authors declare no competing interests.

