## [Peer Review File · Nature Communications]

Reviewers' comments:

Reviewer #1 (NK, innate immunity) (Remarks to the Author):

This is an exciting study which tries to link human NK cell responses to viral infection with NK cell homing to tissues. Although many data sets are solid and persuasive, the one about NK cells in skin blisters is less robust.

This reviewer appreciates that the skin blister experiment is challenging in many ways. However results from samples of 3 patients and 3 HD do not seem to make a robust data set, also because one of the HD behaves much like a patient. Can the Authors address this?

Another major comment is about opting to use Ki67 as the 'master' indicator of response. It does make sense, however this reviewer would like to remind the Authors that NK-cell functions independent of proliferation have made the history of the field (hybrid resistance in irradiated mice). This should at least be discussed by the Authors.

Looking closely at Figure 5B, one sees that IFN- γ responses of CD56^{bright} NK cells to IL-12+IL-18 are close to saturation in any phase of disease, but also in HD. This means that NK cells are poised to respond to IL-12 and IL-18 regardless of the infection. The conclusions about the role of IL-18 must be reassessed in light of this evidence.

What function does Dengue Virus trigger in CD56^{bright} NK cells in in vitro stimulation assays like those in Figure 5?

The results in Figure 6 are a little confusing. The Authors in the text (line 325) clearly list CXCR3, CCR5 and CCR8 as chemokine receptors leading NK cells to move to skin. But then conclude that CD56^{bright} NK cells up regulate CLA, CCR5 (the results show very little up regulation in Fig6a), CXCR3 and CXCR6. How about CCR8? Also in Figure 7 only CLA is measured. What about the other chemokine receptors mentioned before?

But perhaps more worryingly, both Ki67⁺ and Ki67⁻ NK cells up regulate certain chemokine receptors (CCR5, CXCR3, CCR7 and to a lesser degree, CLA - the only receptors specifically unregulated in a small subset of Ki67⁺ CD56^{bright} NK cells is CCR2), therefore invalidating the very premise of using Ki67 an indicator of NK cell response. It seems there is not enough evidence to link responses to migration.

Reviewer #2 (Dengue, NK anti-viral response) (Remarks to the Author):

Zimmer et al (NCOMMS-18-10949):

In this study Zimmer et al have performed a very thorough phenotypic analysis of the NK cell response in the setting of acute and post-febrile Dengue virus infection. Such a study is timely since NK cells are activated and expanded in many viral infections and with the Dengue vaccine field at the stage of Phase III analysis and beyond, the role of NK cells in such settings is of critical importance. The potential role of NK cells either in protection, or possibly the classical immune enhancement clinical phenotype associated with DENV is an important question. The study is well-written, well designed and the data clearly presented in most cases. The well analyzed high-dimensional flow cytometry data is a highlight of the study. However, there are several important points that should be addressed prior to publication:

Major comments:

1. Nowhere in the provided text, either supplementary or in the paper itself, is there any information about where exactly these samples were obtained. This reviewer assumes that it was from a hospital in Singapore? Basic demographics are provided, but no further information. This could be a simple addition to the supplementary text.
2. Was the dengue serotype ever confirmed virologically? If so, this data should be provided as well. In addition, it should be pointed out clearly that all but one of the subjects fell into the Dengue Fever (DF) category with only one DHF subject. This information is critically relevant to readers trying to assess the potential for NK cell involvement in dengue pathogenesis.
3. Were these infections primary, or secondary/tertiary infections? It should be relatively straightforward from any neutralizing antibody profile, whether these may have been primary (or post primary – secondary/tertiary) infections. This is critical to interpretation of the NK cell responses. Since the NK cell profile is primarily immature, or less mature cells (Figures 1, 2, and supplementary Figures 1, 2 data) it would be interesting to interpret this response in light of whether this might be the first, or subsequent exposure of these subjects to DENV. Questions such as – whether a new population NK cells is recruited during each subsequent DENV infection, or perhaps only during acute primary infection? – are left open to interpretation when information about infection (if available), could be presented.
4. The fact that the acute NK response is uncoupled from NK cell education (Figures 3 and sup Fig 3) is important and would lead one to speculate that either NK cell “memory” or activation of already “educated cells” is not involved in DENV responsiveness of NK cells. Again this speculation could be answered by any extant data about the infection history of the subjects. If these are clearly secondary infections then the above speculation would be true. If these are primary infections then the question of whether a new wave or population of NK cells is recruited during secondary infection is still open.
5. This reviewer agrees that the IL-18/IL-18R axis may be at least a part of the reason NK cells are responding to DENV infection. It is already well established that NK cells respond *in vitro* to IL-18 via particular signaling pathways and cytokine production. The acute versus convalescent plasma add back experiment in Figure 4g in particular is compelling, but is a modest increase at best in downstream signaling responsiveness.

Minor comments:

1. The summary of the data Figure 1 (and supplementary Fig 1) is interpreted as showing no real expansion of the relative numbers (percentage) of NK cells within the total NK and T cell lymphocyte population. Expansion and activation of T cells during acute dengue infection is well-documented. It is possible that the NK cells numbers have indeed expanded, but as part of the total lymphocyte subset. This is not a criticism of the study analysis, but is a reality of using cryopreserved PBMC.
2. The authors use a comprehensive panel of assays to determine NK cell responsiveness. Is it in any way surprising that the responsiveness of the NK cells is unchanged in acute infection? While it is true that during chronic and acute HIV-infection, for example, that responsiveness of NK cells is potentially diminished, this finding has not been replicated to as great an extent in other viral infection.
3. The skin blister experiments are a novel assessment of DENV NK cell function and the data is compelling. The pattern of chemokine expression on the circulating NK cells is unique and makes sense given our understanding of DENV biology. It is curious that these cells are present in the periphery and are apparently programmed to re-circulate back to the site (s) of viral replication. One question that troubles me is that DENV replicates at many sites other than the skin – why are NK cells programmed specifically to home back to the skin? Perhaps these are a subset of the total NK cell population and there are other NK populations may be homing elsewhere. This observation does however fit with how we would assume NK cells to infection during an infection. Were the skin blister-derived NK cells also of the CD56^{bright} phenotype, as found in the blood, or were they further differentiated to a CD56^{dim} phenotype. It is not clear if this was (or could be) assessed. The relatively small number of cells obtained from such blisters is acknowledged by the reviewer.

Overall comments:

1. The highlights of this study are the well conducted high-dimensional flow cytometry, the potential

IL-18/IL-18R axis as a mediator of NK cell activation, and the pattern of chemokine expression on the circulating and skin-site NK cells. The data is novel for DENV-infection and is a timely study. An important question that study answers at least in part, is that the response profile of the NK cells is that of a cytokine-driven phenomenon. However, while these findings are of interest to the dengue research community and the NK cell biology communities, the enthusiasm of this reviewer for publication of this paper/data in its current form in Nature Communication is low. The study is still largely a phenotypic survey of NK cells and in vitro assessment of their responsiveness to cytokines. No link is made between IL-18 and its potential source. Do DENV-infected cells express IL-18, for example? Do control subject NK cells recognize DENV-infected target cells any less efficiently than acute or convalescent subject derived NK cells? One critical criterion for publication in this journal is "a paper should represent an advance in understanding likely to influence thinking in the field." While the sampling methods are novel and technical competence is high, the data largely reflects what one might expect to see and is unlikely to dramatically alter thinking in the field.

2. The following experiments would further enhance and potentially elevate this study to Nature communications standards:

a. Do DENV-infected cells express IL-18?

b. Do NK cells, either from control, or infected subjects (acute and convalescent) recognize DENV-infected cells?

c. While transcriptional profiling is not a trivial technique it has become an almost de facto standard and requirement for publications at this level. For example, scRNA-seq would shed a tremendous amount of light on the compelling observations put forward in this study.

Sincerely,

Jeffrey R. Currier, PhD.

(Ms. No. NCOMMS-18-10949)

Point-by-point reply to the reviewers' comments

Reviewer 1

This is an exciting study which tries to link human NK cell responses to viral infection with NK cell homing to tissues. Although many data sets are solid and persuasive, the one about NK cells in skin blisters is less robust.

Author response: We appreciate that the reviewer found our study exciting. Regarding skin blisters, we have now significantly increased our cohort size and performed a more in-depth analysis of skin NK cells in acute DENV infection. It took considerable time to raise this extended cohort, but we now feel that data are more robust and delivers an even clearer message.

This reviewer appreciates that the skin blister experiment is challenging in many ways. However results from samples of 3 patients and 3 HD do not seem to make a robust data set, also because one of the HD behaves much like a patient. Can the Authors address this?

Author response: We have recruited more individuals to the study, which now includes a total of 8 patients with acute DENV infection and 5 healthy controls. Analysis of this extended cohort substantiated our previous finding with the specific appearance of CD69⁺CLA⁺ NK cells in skin of patients with acute DENV but not in controls. We also show that the absolute number of skin-infiltrating NK cells correlates with number of days after symptom debut with the highest number of NK cells present in skin early after symptom debut. We could also show that IL-18, which is increased in circulation of patients with acute DENV, is even further elevated in the skin microenvironment. All of this and additional data are now included in the revised manuscript.

Another major comment is about opting to use Ki67 as the 'master' indicator of response. It does make sense, however this reviewer would like to remind the Authors that NK-cell functions independent of proliferation have made the history of the field (hybrid resistance in irradiated mice). This should at least be discussed by the Authors.

Author response: Proliferation-independent responses, such as cytotoxic or cytokine responses are indeed interesting. For some parts of the analysis, we display the NK cell response in terms of CD69 upregulation in parallel to Ki67 upregulation. In particular, since CD69 and Ki67 expression to a large extent were mutually exclusive on responding NK cells. Regarding hybrid resistance, the authors are very well aware of the early history of the field, including the early work of Bennet and Cudkowicz (e.g., JEM 1971), the finding by Kiessling and Cudkowicz that the phenomena was mediated by NK cells (EJI, 1977), and the findings by Ljunggren (author on the present paper) and Kärre that the phenomenon could be explained by “missing-self” recognition (JEM, 1985; Nature, 1986), i.e. that the elimination of parental grafts to F1 hybrids could be related to cytotoxic rejection responses mediated by NK cells. We have therefore added some sentences relating to the possible role of cytotoxic (rejection) mediated responses mediated by NK cells in context of the clinical responses seen in DF pathogenesis. In this regard, it is interesting to note that Nikzad *et al.* (Science Immunology 2019)

recently showed that NK cells infiltrating skin upon VZV challenge are CD107a⁺ suggestive of an ongoing cytotoxic response.

Looking closely at Figure 5B, one sees that IFN-g responses of CD56bright NK cells to IL-12+IL-18 are close to saturation in any phase of disease, but also in HD. This means that NK cells are poised to respond to IL-12 and IL-18 regardless of the infection. The conclusions about the role of IL-18 must be reassessed in light of this evidence.

Author response: We have performed additional functional experiments using different combinations of non-saturating concentrations of IL-12 and IL-18. This data is now included in Supplementary Figure 6. In line with the original functional data, also at non-saturating levels of these cytokines, the NK cell functional response is at large retained in acute infection as compared to controls. However, we did detect slightly lower levels of IFN γ being produced from both CD56^{bright} and CD56^{dim} NK cells from patients during acute DENV infection in response to stimulation with either IL-12 or IL-18 alone.

Nevertheless, we are not concluding that IL-18 is responsible for altered NK cell functionality but rather that it is linked to the robust proliferative response observed. Furthermore, and as additional support for a role of IL-18 in driving the NK cell response during acute DENV infection, we now show that the skin microenvironment contains even higher levels of IL-18 as compared to plasma in acute DENV infection. In contrast, no change in IL-2, -12, or -15 was observed in skin blister fluid as compared to plasma. A recent *in vitro* study on murine NK cells showed a specific role for IL-18 in driving NK cell proliferation via upregulation of the nutrient transporter CD98, whereas no effect was seen for other NK-cell stimulatory proteins (Almutairi *et al.* JBC 2019). In line with this, we can now also demonstrate that NK cells from patients with acute DENV infection express higher levels of CD98 compared to controls, further substantiating the link between IL-18 and NK cell proliferation during acute DENV infection. All of this is now included in Figure 4.

What function does Dengue Virus trigger in CD56bright NK cells in in vitro stimulation assays like those in Figure 5?

Author response: To address this question, we infected PBMCs with DENV (with or without the chimeric 4G2 mAb that enhances infectivity as well as with UV-inactivated virus as control). Results from these experiments are now included in Figure 6. Both CD56^{bright} and CD56^{dim} NK cells responded with IFN γ , TNF, and MIP-1 β production as well as upregulation of CD69 and TRAIL (Supplementary Figure 7) upon infection.

The results in Figure 6 are a little confusing. The Authors in the text (line 325) clearly list CXCR3, CCR5 and CCR8 as chemokine receptors leading NK cells to move to skin. But then conclude that CD56bright NK cells up regulate CLA, CCR5 (the results show very little up regulation in Fig6a), CXCR3 and CXCR6. How about CCR8? Also in Figure 7 only CLA is measured. What about the other chemokine receptors mentioned before?

Author response: As far as we know, NK cell homing to skin has previously not been studied during acute viral infection in humans. Indeed, previous human studies in chronic viral infection or autoimmunity have indicated CXCR3, CCR5, and CCR8 to be

important. We appreciate the reviewers comment that the histograms in the original version of Figure 6A showed only modest differences whereas the summary data showed a significant upregulation of CLA, CCR5, CXCR6, and CCR9 on responding CD56^{bright} NK cells together with a downregulation of CCR7. Chemokine receptor stainings are challenging to perform, especially on previously frozen PBMC. To substantiate our findings, we have repeated our initial experiments on an additional 5 patients with acute infection as well as on additional community matched controls. These experiments yielded similar results as the original batch of analyzed patients and the complete dataset now contains a sizeable number of acutely infected patients for each chemokine receptor ($n=11-22$ depending on receptor). Altogether, this shows that responding CD56^{bright} NK cells, during acute DENV infection, upregulate CLA, CCR5, CXCR6, and CCR9. A role for CCR5 in skin homing is in line with the previously cited literature for other conditions than acute viral infections. This also makes sense since ligands for CCR5 are produced by keratinocytes in the skin (Ottaviani *et al.* EJI 2006). A role for CXCR6 on NK cells in homing to skin is in line with a recent report assessing NK cells in skin after VZV skin challenge (Nikzad *et al.* Science Immunology 2019).

Regrettably, we have not managed to find an anti-CCR8 mAb that reliably stains human immune cells and have therefore not been able to assess expression of this receptor. We are aware of previous publications on this in relation to skin (for instance McCully *et al.* JI 2018). This is now brought up as a limitation of our study in the discussion.

Regarding skin NK cells in acute DENV infection, and except for CLA, we have now added new data on CXCR3, CCR5, CCR7, and CCR10. A significant fraction of skin NK cells from patients with acute DENV infection expressed CXCR3 and CCR5 and the levels of these receptors were higher than on matched peripheral blood NK cells in all investigated individuals. In contrast, CCR7 and CCR10 was only detected at low levels on skin NK cells.

Collectively, this analysis, together with existing literature, suggests a role for CLA, CXCR3, CCR5, and CXCR6 in NK cell homing to skin during acute DENV infection.

But perhaps more worryingly, both Ki67+ and Ki67- NK cells up regulate certain chemokine receptors (CCR5, CXCR3, CCR7 and to a lesser degree, CLA - the only receptors specifically unregulated in a small subset of Ki67+ CD56bright NK cells is CCR2), therefore invalidating the very premise of using Ki67 an indicator of NK cell response. It seems there is not enough evidence to link responses to migration.

Author response: We appreciate this comment and the fact that the representative stainings in the original submission not clearly depicted the data. As mentioned above, we have now substantiated these findings by repeating them on a second independent cohort. As shown in the heatmap in Figure 7, responding Ki67⁺ (or CD69⁺) NK cells during acute DENV infection are different in their chemokine receptor profile both compared to matched non-responding Ki67⁻ (or CD69⁻) NK cells from the same samples (indicated with * for significance in the figure) as well as compared to healthy controls (indicated with # for significance in the figure). We have now included legends in the figure to indicate what * and # stands for to make it easier to interpret without having to read the figure legend. We have also replaced the representative histograms and the way we depict the representative data so that it more clearly is in line with the

aggregated data. Furthermore, we also show that skin NK cells from patients with acute infection express higher levels of CCR5 and CXCR3 as compared to matched peripheral blood NK cells.

In additional experiments, we also demonstrate that there is an inverse correlation between NK cell numbers per skin blister and day after symptom debut, suggesting an early recruitment of NK cells to the skin during acute DENV infection. It is plausible that this is associated with the increased concentration of IL-18 in skin blister fluid as compared to plasma that we also now find. Indeed, IL-18 has been shown to promote IFN- γ production which in turn led to release of ligands to CCR5 and CXCR3 from keratinocytes in skin (Kanda *et al.* EJI 2007, Ottaviani *et al.* EJI 2006). Altogether, our impression is this new data, as well as reanalysis and better presentation of the original data, now more firmly support our conclusion that NK cells are activated and primed for skin homing during acute infection. In line with past and present new data we have adjusted the writings in the discussion accordingly.

Reviewer 2

In this study Zimmer et al have performed a very thorough phenotypic analysis of the NK cell response in the setting of acute and post-febrile Dengue virus infection. Such a study is timely since NK cells are activated and expanded in many viral infections and with the Dengue vaccine field at the stage of Phase III analysis and beyond, the role of NK cells in such settings is of critical importance. The potential role of NK cells either in protection, or possibly the classical immune enhancement clinical phenotype associated with DENV is an important question. The study is well-written, well designed and the data clearly presented in most cases. The well analyzed high-dimensional flow cytometry data is a highlight of the study. However, there are several important points that should be addressed prior to publication:

Author response: We thank the reviewer for appreciating our study and for finding it timely.

Major comments:

1. Nowhere in the provided text, either supplementary or in the paper itself, is there any information about where exactly these samples were obtained. This reviewer assumes that it was from a hospital in Singapore? Basic demographics are provided, but no further information. This could be a simple addition to the supplementary text.

Author response: All patients were recruited at Tan Tock Seng Hospital in Singapore. This text is now added to the material and methods section of the manuscript.

2. Was the dengue serotype ever confirmed virologically? If so, this data should be provided as well. In addition, it should be pointed out clearly that all but one of the subjects fell into the Dengue Fever (DF) category with only one DHF subject. This information is critically relevant to readers trying to assess the potential for NK cell involvement in dengue pathogenesis.

Author response: The dengue serotype was not confirmed virologically as the early acute blood sample was not available for these patients and DENV is cleared from the blood of infected patients by day 5 of illness onset. All four serotypes of DENV

circulate in Singapore. Patients were enrolled in our study during several years and, interestingly, during this period, the serotype dominance shifted in Singapore. Thus, if we map when patients were recruited in relation to public health data on serotype dominance in Singapore, three time-periods become evident: one with a DENV1-dominance, a second with a DENV2-dominance, and a third period with a mixed DENV1/2-dominance. If we compare the overall NK cell responses and subdivide our cohort into three groups based on serotype dominance, no differences in NK cell responses could be noted. However, with our study design, we could not perform a deeper analysis of tentative differential NK cell responses towards different serotypes of the virus.

Indeed, all investigated subjects (except one) had DF. We now stress this throughout the text. Our aim was to perform a very detailed assessment of NK cell responses in an as well-characterized and homogeneous group of patients as possible. Our results will hopefully function as a platform for future more pathogenesis-oriented studies comparing larger groups of patients with different disease presentations.

3. Were these infections primary, or secondary/tertiary infections? It should be relatively straightforward from any neutralizing antibody profile, whether these may have been primary (or post primary – secondary/tertiary) infections. This is critical to interpretation of the NK cell responses. Since the NK cell profile is primarily immature, or less mature cells (Figures 1, 2, and supplementary Figures 1, 2 data) it would be interesting to interpret this response in light of whether this might be the first, or subsequent exposure of these subjects to DENV. Questions such as – whether a new population NK cells is recruited during each subsequent DENV infection, or perhaps only during acute primary infection? – are left open to interpretation when information about infection (if available), could be presented.

Author response: Based on the rapid tests performed at the clinic at the time of patient recruitment during the acute phase of infection, we have performed a new subgroup-analysis comparing NK cell responses to primary vs. secondary DENV infection. This is now included as new data in Figure 3. When assessing Ki67 and CD69 expression in CD56^{bright} and CD56^{dim} NK cells, no differences comparing primary vs. secondary DENV infection could be noted. See also our answer to the comment below, where we further elaborate on NK cell “memory” in acute DENV infection. Collectively, our results indicate that NK cell “memory” does not develop in response to DENV.

4. The fact that the acute NK response is uncoupled from NK cell education (Figures 3 and sup Fig 3) is important and would lead one to speculate that either NK cell “memory” or activation of already “educated cells” is not involved in DENV responsiveness of NK cells. Again this speculation could be answered by any extant data about the infection history of the subjects. If these are clearly secondary infections then the above speculation would be true. If these are primary infections then the question of whether a new wave or population of NK cells is recruited during secondary infection is still open.

Author response: This is a very relevant comment, especially given the recent publication by Nikzad *et al.* (Science Immunology 2019), showing adaptive immunity to certain viral antigens by human NK cells (HIV and VZV). To bring further clarity to this, we have increased the cohort of patients with acute infection and have assessed NK cell responses in relation to education status (16 patients are studied compared to nine patients in the original analysis). In this extended analysis, similar results were obtained

as in our initial analysis, suggesting that NK cells respond independently of NK cell education. Furthermore, we identified two patients with preexisting “adaptive-like” NK cell expansions (high expression of NKG2C and CD57). When specifically studying the response pattern of these “adaptive-like” NK cells during acute DENV infection, we did not notice preferential responses occurring within the adaptive-like NK cell population. Rather, these cells responded in a similar fashion as non-adaptive NK cells. Furthermore, since “adaptive-like” NK cells typically are educated (Björkström *et al.* JEM 2011, Beziat *et al.* Blood 2013), and if these cells would specifically contribute to the NK cell response, we would have observed that in the analysis of responses based on education status. Finally, and as pointed out above, we have also undertaken a subgroup-analysis of patients with primary *vs.* secondary DENV infection. This showed that NK cell response occurs in a similar fashion independent of infection history. Altogether, this would suggest that NK cell “memory” does not develop in response DENV.

5. This reviewer agrees that the IL-18/IL-18R axis may be at least a part of the reason NK cells are responding to DENV infection. It is already well established that NK cells respond in vitro to IL-18 via particular signaling pathways and cytokine production. The acute versus convalescent plasma add back experiment in Figure 4g in particular is compelling, but is a modest increase at best in downstream signaling responsiveness.

Author response: We appreciate the reviewers comment. To further substantiate a role for IL-18, we have now obtained new data showing that IL-18 is also present in skin blister fluid at a concentration even higher than that in plasma during acute infection. In contrast, no changes in IL-2, -12, or -15 were observed in skin blister fluid as compared plasma. A recent *in vitro* study on murine NK cells showed a specific role for IL-18 in driving NK cell proliferation via upregulation of the nutrient transporter CD98, whereas no effect on CD98 upregulation was seen for other NK-cell stimulatory cytokines (Almutairi *et al.* JBC 2019). In line with this, we can now also demonstrate that NK cells from patients with acute DENV infection express higher levels of CD98 compared to controls. This data is now included in Figure 4 and further strengthens the link between IL-18 and the NK cell proliferative response in acute DENV infection.

Minor comments:

1. The summary of the data Figure 1 (and supplementary Fig 1) is interpreted as showing no real expansion of the relative numbers (percentage) of NK cells within the total NK and T cell lymphocyte population. Expansion and activation of T cells during acute dengue infection is well-documented. It is possible that the NK cells numbers have indeed expanded, but as part of the total lymphocyte subset. This is not a criticism of the study analysis, but is a reality of using cryopreserved PBMC.

Author response: We agree with the reviewer. This is also what we see when assessing frequency of NK cells out of total lymphocytes in other infections; i.e., that it is very often stable in peripheral blood and thus the absolute numbers of cells would here add an additional level (see for instance Zimmer and Björkström JID 2018 or Strunz and Björkström Nature Communication 2018 for studies on NK cells in HBV and HCV infections). Interestingly, in expanding our skin blister sample size, we now see an inverse correlation between NK cell numbers per skin blister and number of days after

symptom debut. This data suggests active recruitment of NK cells to the skin early in infection (see also answer to comment below).

2. The authors use a comprehensive panel of assays to determine NK cell responsiveness. Is it in any way surprising that the responsiveness of the NK cells is unchanged in acute infection? While it is true that during chronic and acute HIV-infection, for example, that responsiveness of NK cells is potentially diminished, this finding has not been replicated to as great an extent in other viral infection.

Author response: We agree with the reviewer that this is somewhat puzzling. In our previous work, we have in certain infections, for instance reactivation of chronic HBV-infection, seen enhanced NK cell responses (Zimmer *et al.* JID 2018). In other studies, we have instead observed that the viral infection only has a limited impact on NK cell responsiveness (chronic HCV infection, Strunz *et al.* Nature Communications 2018). In new experiments using non-saturating amounts of IL-12 or IL-18, we show that NK cells from patients with acute infection present with a slight, yet significant, functional defect, especially with respect to IFN- γ production. These new data are now included in supplementary Figure 6, however since these differences were minor we are not strongly emphasizing these results.

3. The skin blister experiments are a novel assessment of DENV NK cell function and the data is compelling. The pattern of chemokine expression on the circulating NK cells is unique and makes sense given our understanding of DENV biology. It is curious that these cells are present in the periphery and are apparently programmed to re-circulate back to the site (s) of viral replication. One question that troubles me is that DENV replicates at many sites other than the skin – why are NK cells programmed specifically to home back to the skin? Perhaps these are a subset of the total NK cell population and there are other NK populations may be homing elsewhere. This observation does however fit with how we would assume NK cells to infection during an infection. Were the skin blister-derived NK cells also of the CD56^{bright} phenotype, as found in the blood, or were they further differentiated to a CD56^{dim} phenotype. It is not clear if this was (or could be) assessed. The relatively small number of cells obtained from such blisters is acknowledged by the reviewer.

Author response: We have not had the possibility to assess NK cells in other anatomical locations than skin during acute infection. For instance, the gut or liver would indeed have been interesting to study. The upregulation of CXCR6 and CCR9 that we observe would most likely also promote homing to liver and gut respectively. Regarding skin NK cells, we have now significantly expanded our analysis. We now show that there is a preferential accumulation of CD56^{bright} NK cells in skin. We also find an inverse correlation between numbers of NK cells per skin blister and days after symptom debut, suggesting early recruitment of NK cells to the skin. We also show that the CLA⁺CD69⁺ NK cell subset that we found in skin is only present during acute DENV infection and not in healthy controls. Finally, NK cells recruited to skin expressed higher levels of CXCR3 and CCR5 during acute infection, which is in line with the chemokine-receptor modulation observed in circulation.

Overall comments:

1. The highlights of this study are the well conducted high-dimensional flow cytometry, the potential IL-18/IL-18R axis as a mediator of NK cell activation, and the pattern of chemokine

expression on the circulating and skin-site NK cells. The data is novel for DENV-infection and is a timely study. An important question that study answers at least in part, is that the response profile of the NK cells is that of a cytokine-driven phenomenon. However, while these findings are of interest to the dengue research community and the NK cell biology communities, the enthusiasm of this reviewer for publication of this paper/data in its current form in Nature Communication is low. The study is still largely a phenotypic survey of NK cells and in vitro assessment of their responsiveness to cytokines. No link is made between IL-18 and its potential source. Do DENV-infected cells express IL-18, for example? Do control subject NK cells recognize DENV-infected target cells any less efficiently than acute or convalescent subject derived NK cells? One critical criterion for publication in this journal is “a paper should represent an advance in understanding likely to influence thinking in the field.” While the sampling methods are novel and technical competence is high, the data largely reflects what one might expect to see and is unlikely to dramatically alter thinking in the field.

Author response: We appreciate that the reviewer finds our study timely, of high quality, and interesting both to the dengue research environment as well as the NK cell biology community. We have in the revised version of the manuscript added significant new data including: i) a further dissection into potential “memory” NK cell responses where the data suggests that these responses are not present in secondary DENV infection, ii) a more detailed assessment of the IL-18 link where we now show that IL-18 also is specifically increased within the skin microenvironment and that this together with upregulation of CD98 on NK cells suggests a possible mechanism for the proliferative NK cell response, and finally iii) a much more robust assessment of skin NK cells in patients with acute DENV, where our findings for the first time, to the best of our knowledge, demonstrate that human NK cells localize to the site of infection during an acute viral infection.

Collectively, we do believe that our manuscript, with the new data added in the revision in response to the reviewer comments, represent significant advances both within the NK cell and the dengue research fields.

2. *The following experiments would further enhance and potentially elevate this study to Nature communications standards:*

a. *Do DENV-infected cells express IL-18?*

Author response: We have performed new experiments to address this. DENV infection of PBMC (with or without the chimeric 4G2 mAb that enhances infectivity) did not yield IL-18 production but instead resulted in production of IFN α and TNF. On the other hand, we managed to measure IL-18 in skin blister fluid from patients with acute infection. This showed that the IL-18 levels are even further increased within the skin microenvironment on top of the already elevated levels in plasma. No differences were observed for IL-2, -12, or -15 in skin blister vs plasma from patients with acute infection. Altogether, this would suggest that immune cells found in circulation are not the direct source of IL-18 during acute DENV infection but rather that either parenchymal cells of infected organs (for instance keratinocytes which has been shown in numerous papers) or tissue-resident immune cells (such as macrophages or Langerhans cells) represent the source of IL-18. As a further link between IL-18 and NK cell proliferation, we also show that NK cells upregulate the nutrient transporter CD98 during acute infection (see discussion above). This is in line with a recent paper

on mouse NK cells revealing a new mTOR-independent mechanism for NK cell proliferation via IL-18 and upregulation of CD98.

b. Do NK cells, either from control, or infected subjects (acute and convalescent) recognize DENV-infected cells?

Author response: We now include new data showing that both CD56^{bright} and CD56^{dim} NK cells respond during DENV infection of PBMC (Figure 6). This response was further enhanced when the level of infectivity was increased using the 4G2 mAb. This data is now shown together with the other functional data.

c. While transcriptional profiling is not a trivial technique it has become an almost de facto standard and requirement for publications at this level. For example, scRNA-seq would shed a tremendous amount of light on the compelling observations put forward in this study.

Author response: The vast majority of our experiments for this manuscript have been performed at Karolinska Institutet in Sweden where DENV is classified as a BSL-3 pathogen. For us to FACS sort subsets of NK cells for scRNA-seq would require access to a BSL-3 FACS sorter which unfortunately does not exist in Sweden. Although interesting, we would argue that the proposed analysis is beyond the scope of the current manuscript and would represent a complete story by itself.

REVIEWERS' COMMENTS:

Reviewer #1 (Remarks to the Author):

The Authors have adequately responded to the points raised in my review

Reviewer #2 (Remarks to the Author):

Zimmer et al (NCOMMS-18-10949A): Second round review.

This second round review is in response to the revised study of Zimmer et al. who have performed a very thorough phenotypic analysis of the NK cell response in the setting of acute and post-febrile Dengue virus infection. Several important questions were raised in the first round reviews, however the authors have made substantive and important changes to their manuscript which has improved this exciting study. The study advances the field by performing one of the first in-depth characterizations of NK cell responses during acute DENV infection, including direct ex vivo analysis of skin blister fluid. The authors have addressed all of my previous major concerns. This reviewer believes the study meets Nature Communications standards provided the authors can respond to the following remaining questions.

Comments based upon previous review questions:

Major comments:

1. The authors have increased the sample size significantly. 32 patients and 26 control subjects provides a sample set appropriate for the conclusions. This reviewer appreciates the inclusion of the study site data and concise patient demographic information. Especially important is the increase in skin blister fluid sampling to 13 total subjects. This data is much more robust. The addition of specific statistical tests applied to each dataset in the appropriate figure legend is also appreciated. The activation marker data for circulating NK cells is impressive.
 - a. The one point that should still be made is that from Figures 1A-D it is clear that Ki67 and CD69 are not-co-expressed on the same cells. Is that true in general? If so, this point, although minor in the overall study, should be made clear. The CD38 analysis is presumably the MFI on all NK cells? A simple acknowledgement of this in the Results section describing this data would be sufficient.
2. I thank the authors for the detailed response to my question concerning the DENV serotype, virological assessment and confirmation of DENV-infection. This is now clear.
3. The new analysis concerning the primary versus secondary (or post-primary) infection status stratification of the subjects strengthens this paper for the dengue researcher readership. While it is clear that the NK cell response is independent of primary versus secondary infection status, this is important to know in the context dengue pathobiology.
4. Following from the response to my concerns raised in point three above, the authors have proved sufficiently to me that the NK cell response to DENV is independent of their education status. Again, this analysis strengthens the paper significantly.
5. The inclusion of the IL-18 and IL-18R studies on NK cells and in the skin blister fluid is particularly exciting. The increases in IL-18 post-infection in plasma and skin during acute infection is intriguing and makes sense in the context of overall NK cell activation during acute dengue.

Minor comments:

1. The reviewer appreciates the inclusion of the chemokine receptor analysis. While I was originally concerned about the fact that DENV replicates at many sites other than the skin – the CCR-based flow

cytometry is consistent with both skin-homing and homing to other potential sites of infection. The ex vivo analysis of NK cells from skin blisters is the most exciting and novel part of this study. All of the phenotyping data is consistent with increased NK cell presence in skin during the acute phase of dengue.

a. The authors should clearly point out the clear disparity between CLA⁺/CD69⁺ NK cells in skin versus blood. This observation is clearly in alignment with the overall message of the study: recently activated (CD69⁺) and skin tissue infiltrating NK cells (CD69⁺ and CLA⁺) are increased during dengue infection. Linking the observed responses in the blood to specific migration to tissues, such as skin, will be important to readers.

Final comments: This study is an exciting analysis of NK cells in the setting of DENV infection fills a current void in the field – do NK cells infiltrate the skin, an important initial site of DENV entry into the body, and what drives this response.

Sincerely,

Jeffrey R. Currier

(Ms. No. NCOMMS-18-10949B)

Point-by-point reply to the reviewers' comments

Reviewer #1

The Authors have adequately responded to the points raised in my review

Author response: We thank reviewer #1 for the positive feedback.

Reviewer #2

Major comments:

1. The authors have increased the sample size significantly. 32 patients and 26 control subjects provides a sample set appropriate for the conclusions. This reviewer appreciates the inclusion of the study site data and concise patient demographic information. Especially important is the increase in skin blister fluid sampling to 13 total subjects. This data is much more robust. The addition of specific statistical tests applied to each dataset in the appropriate figure legend is also appreciated. The activation marker data for circulating NK cells is impressive.

a. The one point that should still be made is that from Figures 1A-D it is clear that Ki67 and CD69 are not-co-expressed on the same cells. Is that true in general? If so, this point, although minor in the overall study, should be made clear. The CD38 analysis is presumably the MFI on all NK cells? A simple acknowledgement of this in the Results section describing this data would be sufficient.

Author response: We thank the reviewer for pointing out that Ki67 and CD69 co-expression is largely mutually exclusive. To make this even more clear, we added text to the results section as well as to the discussion. This observation is similar to what Marquardt et al. (J Immunol. 2015) could show for the NK cell after yellow fever virus vaccination. The mutually exclusive expression pattern is indicative of cells being in different phases of the response. CD69 is upregulated early upon activation whereas Ki67 is upregulated later during the response.

2. I thank the authors for the detailed response to my question concerning the DENV serotype, virological assessment and confirmation of DENV-infection. This is now clear.

Author response: We are pleased to hear that this has become clearer now.

3. The new analysis concerning the primary versus secondary (or post-primary) infection status stratification of the subjects strengthens this paper for the dengue researcher readership. While it is clear that the NK cell response is independent of primary versus secondary infection status, this is important to know in the context dengue pathobiology.

Author response: We agree with the reviewer and appreciate this comment.

4. *Following from the response to my concerns raised in point three above, the authors have proved sufficiently to me that the NK cell response to DENV is independent of their education status. Again, this analysis strengthens the paper significantly.*

Author response: We are happy to hear that the inclusion of additional donors strengthened the analysis in order to prove that the NK cell response occurs independently from education.

5. *The inclusion of the IL-18 and IL-18R studies on NK cells and in the skin blister fluid is particularly exciting. The increases in IL-18 post-infection in plasma and skin during acute infection is intriguing and makes sense in the context of overall NK cell activation during acute dengue.*

Author response: We absolutely agree with the reviewer.

Minor comments:

1. *The reviewer appreciates the inclusion of the chemokine receptor analysis. While I was originally concerned about the fact that DENV replicates at many sites other than the skin – the CCR-based flow cytometry is consistent with both skin-homing and homing to other potential sites of infection. The ex vivo analysis of NK cells from skin blisters is the most exciting and novel part of this study. All of the phenotyping data is consistent with increased NK cell presence in skin during the acute phase of dengue.*

Author response: We greatly appreciate the reviewer's comment and agree that the analysis of NK cells from skin blister fluid is very exciting and supporting our initial findings.

a. *The authors should clearly point out the clear disparity between CLA+/CD69+ NK cells in skin versus blood. This observation is clearly in alignment with the overall message of the study: recently activated (CD69+) and skin tissue infiltrating NK cells (CD69+ and CLA+) are increased during dengue infection. Linking the observed responses in the blood to specific migration to tissues, such as skin, will be important to readers.*

Author response: We agree with the reviewer that this is an important point and added text to the results section and the discussion.

Final comments: This study is an exciting analysis of NK cells in the setting of DENV infection fills a current void in the field – do NK cells infiltrate the skin, an important initial site of DENV entry into the body, and what drives this response.

Author response: We are happy that the reviewer shares our excitement concerning the findings in our study.